# Functional membrane microdomains and the hydroxamate siderophore transporter ATPase FhuC govern Isd-dependent heme acquisition in *Staphylococcus aureus*

Lea Antje Adolf[1,2,3], Angelika Müller-Jochim[1,2,3], Lara Kricks[4,5,6], Jan-Samuel Puls[7], Daniel Lopez[4,5,6], Fabian Grein[7,8], Simon Heilbronner[1,2,3,9,10]*

[1]Department of Infection Biology, Interfaculty Institute of Microbiology and Infection Medicine, University of Tübingen, Tübingen, Germany; [2]Cluster of Excellence EXC 2124 Controlling Microbes to Fight Infections, Tübingen, Germany; [3]Interfaculty Institute of Microbiology and Infection Medicine, Institute for Medical Microbiology and Hygiene, UKT Tübingen, Tübingen, Germany; [4]National Centre for Biotechnology, Spanish National Research Council (CNB-CSIC), Madrid, Spain; [5]Research Centre for Infectious Diseases (ZINF), University of Würzburg, Würzburg, Germany; [6]Institute for Molecular Infection Biology (IMIB), University of Würzburg, Würzburg, Germany; [7]Institute for Pharmaceutical Microbiology, University Hospital Bonn, University of Bonn, Bonn, Germany; [8]German Center for Infection Research (DZIF), partner site Bonn-Cologne, Bonn, Germany; [9]German Center for Infection Research (DZIF), Tübingen, Germany; [10]Faculty of Biology: Microbiology, Ludwig-Maximilians-Universität München, München, Germany

*For correspondence:
simon.heilbronner@bio.lmu.de

**Abstract** Sufficient access to transition metals such as iron is essential for bacterial proliferation and their active limitation within host tissues effectively restricts infection. To overcome iron limitation, the invasive pathogen *Staphylococcus aureus* uses the iron-regulated surface determinant (Isd) system to acquire hemoglobin-derived heme. While heme transport over the cell wall is well understood, its transport over the membrane is hardly investigated. In this study, we show the heme-specific permease IsdF to be energized by the general ATPase FhuC. Additionally, we show that IsdF needs appropriate location within the membrane for functionality. The membrane of *S. aureus* possesses special compartments (functional membrane microdomains [FMMs]) to organize membrane complexes. We show IsdF to be associated with FMMs, to directly interact with the FMM scaffolding protein flotillin A (FloA) and to co-localize with the latter on intact bacterial cells. Additionally, Isd-dependent bacterial growth required FMMs and FloA. Our study shows that Isd-dependent heme acquisition requires a highly structured cell envelope to allow coordinated transport over the cell wall and membrane and it gives the first example of a bacterial nutrient acquisition system that depends on FMMs.

## Editor's evaluation

In this fundamental manuscript, the authors provide compelling evidence that a housekeeping ATPase is required for heme utilization in the important pathogen *Staphylococcus aureus* through its interaction with the canonical heme transporter in this organism. The authors convincingly show that this complex associates with functional membrane microdomains and thus establishes a new paradigm for regional localization of the heme transport system in the staphylococci.

The work will be of interest to microbiologists, particularly those studying transport for macromolecules.

## Introduction

Transition metals such as iron, manganese, copper, and zinc are essential trace elements for all kingdoms of life. Due to their redox potential, transition metals can convert between divalent and trivalent states making them ideal to support enzymatic processes. Molecular iron is of major importance in this regard as it is essential for several central metabolic and cellular processes including glycolysis, oxidative decarboxylation, respiration, and DNA replication (*Schaible and Kaufmann, 2004*). Accordingly, appropriate acquisition of iron is essential for bacterial growth and this dependency is a major target for eukaryotic innate immune strategies to limit bacterial infections. Targeted depletion of human body fluids from trace metals effectively limits bacterial proliferation and is referred to as nutritional immunity (*Murdoch and Skaar, 2022*; *Weinberg, 1975*). To overcome host-induced iron starvation, bacterial pathogens have developed a range of strategies to acquire iron during infection (*Sheldon et al., 2016*). They can secrete siderophores, which possess sufficient affinity toward $Fe^{3+}$ to extract the ion from host-chelating molecules (e.g. transferrin, lactoferrin) (*Miethke and Marahiel, 2007*). *Staphylococcus aureus* secretes the carboxylate-type siderophores staphyloferrin A (SA) and staphyloferrin B (SB), and their iron-saturated forms are acquired by the membrane-located HtsABC and SirABC systems, respectively (*Beasley et al., 2009*; *Cheung et al., 2009*). Additionally, *S. aureus* expresses systems for acquisition of siderophores produced by other bacterial species (xenosiderophores). The $FhuBGCD_1D_2$ system enables acquisition of hydroxamate-type siderophores (*Sebulsky and Heinrichs, 2001*) and the SstABCD system allows acquisition of iron-chelating molecules of the catecholate-type including xenosiderophores, but also human-derived catecholamine stress hormones like epinephrine, norepinephrine, or dopamine (*Beasley et al., 2011*). Moreover, the FeoAB transporter presumably allows acquisition of inorganic $Fe^{2+}$(*Sheldon and Heinrichs, 2015*). In humans, much of the iron is bound to heme in hemoglobin (Hb), which can be released from erythrocytes by the action of cytolytic toxins (*Cassat and Skaar, 2013*; *Spaan et al., 2015*). A well-known mechanism for heme acquisition is the iron-regulated surface determinant (Isd) systems. This was first described for the invasive pathogen *S. aureus* (*Mazmanian et al., 2003*). Similar systems have since been identified in *Staphylococcus lugdunensis* (*Heilbronner et al., 2016*), *Staphylococcus capitis,* and *Staphylococcus caprea* (*Sun et al., 2020*) as well as in phylogenetically more distant Gram-positive pathogens such as *Bacillus anthracis* (*Gat et al., 2008*; *Maresso et al., 2006*), *Bacillus cereus* (*Abi-Khalil et al., 2015*), and *Listeria monocytogenes* (*Klebba et al., 2012*; *Xiao et al., 2011*). All Isd systems contain surface-anchored molecules to extract heme from host hemoproteins and to guide it over the cell wall to a membrane-located ATP-binding cassette (ABC) transporter. In the cytosol, the heme is degraded to release ionic iron. The Isd system of *S. aureus* is best studied (*Sheldon and Heinrichs, 2015*). Here, the cell wall-anchored proteins (CWAs) IsdB and IsdH bind Hb and Hb-haptoglobin complexes, respectively. Heme is removed from the hemoproteins by IsdA and IsdB and transferred to IsdC in the cell wall. Heme is then transferred to the heme-specific lipoprotein IsdE on the outer leaflet of the cell membrane. The permease IsdF functions as homodimer and transports heme into the cytosol. There, the two monooxygenases IsdG and IsdI release iron from heme.

The passage of heme across the cell wall of *S. aureus* has been studied in great detail. In contrast, the process of heme transport across the membrane remains somewhat undefined. Importantly, an ATPase is not encoded within the *isd* operon of *S. aureus*, raising the question of how the transport of heme is energized. Additionally, it is unclear if heme funneling across the cell wall demands the membrane transporter to be in a specific location within the liquid mosaic of the membrane to ensure effective passage of heme from the CWAs to the lipoprotein IsdE. It is increasingly recognized that bacterial membranes represent highly structured cellular compartments. In this regard, functional membrane microdomains (FMMs) have gained increasing attention in the recent years (*Bach and Bramkamp, 2013*; *Bramkamp and Lopez, 2015*; *Farnoud et al., 2015*; *López and Kolter, 2010*; *Matsumoto et al., 2006*; *Yokoyama and Matsui, 2020*). FMMs are specific domains in the bacterial membrane that, in the case of *S. aureus*, consist of the polyisoprenoid staphyloxanthin and its derivative lipids. An intrinsic characteristic of FMMs is the structural protein flotillin A (FloA) recruiting proteins to the FMMs and promoting their oligomerization (*Bramkamp and Lopez, 2015*; *López*

*and Kolter, 2010*). FMMs have been shown to be crucial to coordinate diverse cellular functions in *S. aureus*. On one hand, they control activity of cell wall biosynthetic enzymes such as PBP2a and are essential for expression of resistance to β-lactam antibiotics in methicillin-resistant *S. aureus* (MRSA) (*García-Fernández et al., 2017*). On the other hand, several membrane-associated processes such as the secretion of type VII secretion effector proteins (*Mielich-Süss et al., 2017*) or the RNase Rny depend on FloA (*Koch et al., 2017*).

In this study, we have used bacterial two-hybrid assays to show direct interaction between the iron-responsive ATPase FhuC and the heme permease protein IsdF. Additionally, we have shown that Hb-dependent growth requires FhuC activity, suggesting that FhuC energizes heme transport in *S. aureus*. Moreover, the heme permease IsdF was shown to be associated with FMMs, to interact directly with FloA and to co-localize with the latter in the bacterial envelope. Accordingly, Hb-dependent proliferation was reliant on the correct formation of FMMs. Finally, we demonstrated that appropriate FMM formation is also required for Isd-dependent proliferation of *S. lugdunensis*, suggesting that FMM-dependent structuring of the membrane is crucial for the functionality of Isd systems in staphylococci.

## Results

### FhuC is crucial for heme-dependent proliferation of *S. aureus*

The Isd system facilitates the acquisition of heme from Hb. Heme membrane transport involves an ABC transporter consisting of the heme-specific lipoprotein (IsdE) and the transmembrane protein

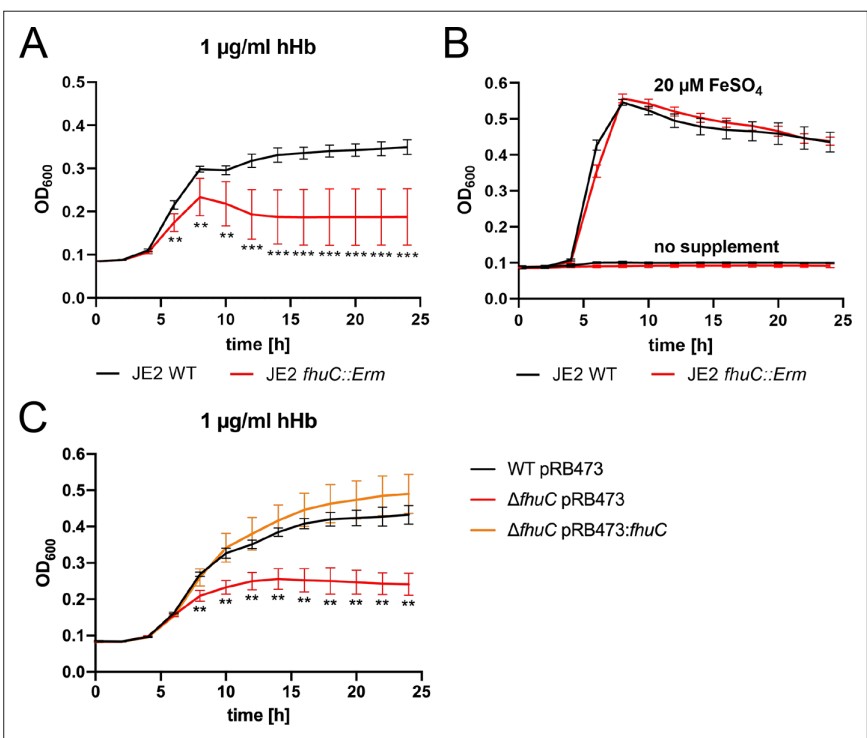

**Figure 1.** FhuC is needed for hemoglobin (Hb)-dependent proliferation of *S. aureus*. One-hundred μl (**A, B**) or 500 μl (**C**) of bacterial cultures were grown with human Hb or $FeSO_4$ as the sole source of iron in 96-well (**A, B**) or 48-well (**C**) format and $OD_{600}$ was monitored over time. For reasons of clarity, values taken every 2 hr are displayed. (**A, B**) *S. aureus* USA300 JE2 wild type (WT) and USA300 JE2 *fhuC::Erm*. Means and SD of six experiments are shown. (**C**) *S. aureus* Newman Δ*fhuC* was complemented using a plasmid expressing FhuC from the native promotor (pRB473:*fhuC*). Newman WT pRB473, Δ*fhuC* pRB473, and Δ*fhuC* pRB473:*fhuC*. Means and SD of three experiments are shown. (**A, C**) Statistical analysis comparing the WT strains and the *fhuC* mutants was performed using GraphPad Prism 9 Student's unpaired t-test. \*\*p<0.01, \*\*\*p<0.001.

The online version of this article includes the following source data for figure 1:

**Source data 1.** FhuC is needed for hemoglobin (Hb)-dependent proliferation of *S. aureus*.

IsdF (*Grigg et al., 2007*; *Mazmanian et al., 2003*). Interestingly, an ATPase to energize membrane transport is not encoded within the *S. aureus isd* operon. Similarly, the operons encoding the SA and SB membrane transport systems (HtsABC and SirABC) lack intrinsic ATPases and both systems were previously shown to be energized by FhuC (*Beasley et al., 2009*; *Speziali et al., 2006*). This ATPase is encoded within the *fhuCBG* operon allowing hydroxamate siderophore transport. Thus, FhuC appears to act as a housekeeping ATPase for iron acquisition in *S. aureus*. In order to determine if FhuC is required for heme acquisition by Isd, we studied growth of *fhuC* mutants in iron-limited medium.

We found that a *S. aureus* USA300 JE2 *fhuC::Erm* mutant, obtained from the Nebraska transposon mutant library (*Fey et al., 2013*), showed significantly reduced growth when human Hb (hHb) was supplied as a sole source of nutrient iron (*Figure 1A*). This deficit was rescued by addition of iron(II) sulfate (*Figure 1B*), suggesting that FhuC energizes heme transport across the membrane. To investigate this further, we created an isogenic deletion mutant of *fhuC* (ΔfhuC) in *S. aureus* strain Newman. Again, the *fhuC*-deficient strain showed a pronounced growth deficit in the presence of Hb. Importantly, plasmid-based expression of FhuC restored Hb-dependent growth (*Figure 1C*). This supports the idea that FhuC is needed for Isd-dependent heme acquisition.

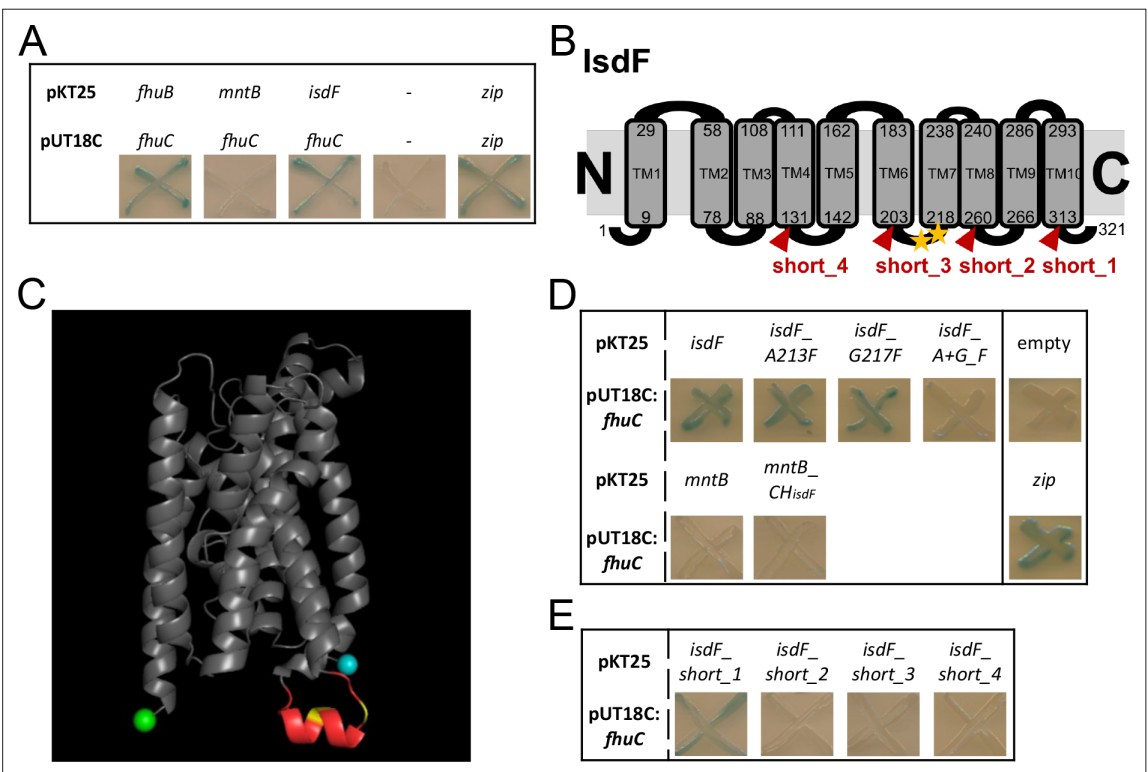

**Figure 2.** FhuC interacts directly with IsdF. (**A, D, E**) *Escherichia coli* BTH101 was co-transformed with pUT18C:*fhuC* and pKT25 vectors expressing permeases of interest. Where protein-protein interactions occur, the T25 and T18 catalytic domains of adenylate cyclase dimerize forming an active enzyme which produces cAMP. This activates LacZ expression leading to X-Gal degradation and blue signals on indicator plates. (**A**) Positive control, leucine zippers (zip). Negative control, empty vectors (pKT25+pUT18C) (-). (**B**) Schematic representation of the IsdF topology prediction using TOPCONS. The amino acids marking the transmembrane domains (TM) are shown with the conserved A213 and G217 indicated by yellow asterisks. Truncations are indicated in red. (**C**) IsdF structure prediction using Alphafold with visualization using PyMOL. The coupling helix is shown in red with the conserved A231 and G217 in yellow. The N-terminus is in cyan and the C-terminus is in green. (**D**) Bacterial adenylate cyclase two-hybrid (BACTH) analysis of FhuC and IsdF with single and double amino acid substitutions as well as with the MntB_CH$_{isdF}$ fusion protein (coupling helix IsdF). (**E**) BACTH analysis of FhuC and truncated IsdF derivatives.

The online version of this article includes the following source data and figure supplement(s) for figure 2:

**Source data 1.** FhuC interacts directly with IsdF.

**Figure supplement 1.** FhuC interaction with iron permeases and topology predictions.

**Figure supplement 1—source data 1.** FhuC interaction with iron permeases and topology predictions.

## Conserved amino acids in the coupling helix of IsdF promote the interaction with FhuC

We used the bacterial adenylate cyclase two-hybrid system (BACTH) to determine whether FhuC physically interacts with the Isd permease IsdF. In BACTH, putatively interacting proteins are fused to domains of an adenylate cyclase. Interaction of the two proteins reconstitutes adenylate cyclase which allows expression of β-galactosidase (LacZ). Co-expression of FhuC with its native permease FhuB as well as with IsdF generated distinct positive signals (*Figure 2A*). In contrast, co-expression of FhuC with the iron-independent permease MntB (manganese uptake) did not result in detectable LacZ activity (*Figure 2A*). Quantification of LacZ activity confirmed the plate-based screening (*Figure 2—figure supplement 1A*). This suggests that FhuC energizes the heme transport system IsdEF and structural characteristics within the permease must allow discrimination of FhuC targets and non-targets.

ATPases and their respective permeases display Q-loops and coupling helixes, respectively, to mediate their interaction (*Hollenstein et al., 2007*; *Wen and Tajkhorshid, 2011*). We speculated that conserved motifs within the coupling helixes of iron compound permeases might promote their energization by FhuC. To investigate this, we modeled the topology of the permease IsdF using TOPCONS (*Tsirigos et al., 2015*) and Alphafold (*Jumper et al., 2021*; *Varadi et al., 2022*). This resulted in prediction of 10 transmembrane helices. Both the N- and C-termini of the protein as well as four loops were proposed to be located in the cytoplasm (*Figure 2B*, *Figure 2—figure supplement 1B*). The third cytoplasmic loop (aa 204–217) contained the cytosolic coupling helix (*Figure 2B+C*). To identify motifs that might determine energization by FhuC, we aligned the sequences of the coupling helixes of all putative FhuC interaction partners (IsdF, FhuB, FhuG, SirB, SirC, HtsB, HtsC), as well as of MntB, using Clustal Omega (*Madeira et al., 2022*). This analysis showed that within all putative coupling helixes only a single glycine residue (G217 in IsdF) is conserved. Additionally, a single alanine residue (A213 in IsdF) was conserved in all iron compound permeases but not in MntB (*Figure 2B+C*, *Figure 2—figure supplement 1C*). Many coupling helixes contain an EAAxxxGxxxxxxxxxIxLP (EAA) motif (*ter Beek et al., 2014*). We speculated that the AxxxG motif identified in the iron permeases might be important to mediate interaction with FhuC. Exchange of the alanine 213 to the bulky amino acid phenylalanine (A213F) did not prevent interaction with FhuC. The same was true when glycine 217 was exchanged for phenylalanine (G217F). However, combination of both substitutions abrogated the interaction (*Figure 2D*). These results show the importance of A213 and G217 for the recognition of FhuC, but they do not explain how MntB and IsdF are discriminated. To examine this, we inserted the full-length coupling helix of IsdF at the appropriate position within MntB. Interestingly, the resulting fusion protein remained negative in BACTH analysis with FhuC (*Figure 2D*), suggesting that the sequence and structure of the coupling helix alone is not sufficient to allow FhuC to discriminate between IsdF and MntB. To further investigate this, we created a series of C-terminal truncations of IsdF (*Figure 2B+E*). Deletion of the C-terminal cytoplasmic domain (short_1) did not affect the interaction with FhuC. However, truncation at the fourth cytosolic loop (short_2) abrogated the interaction and the same was true for additional truncations lacking more extended C-terminal fragments (short_3/short_4) (*Figure 2E*). These data indicate that besides the coupling helix, the C-terminal part of IsdF is crucial for appropriate formation of the IsdF-FhuC complex.

## IsdF is located within FMMs and interacts with FloA

FMMs and the associated structuring protein FloA promote the correct structure of membrane-associated proteins and protein complexes in *S. aureus* (*Bramkamp and Lopez, 2015*). Proteomic profiles of FMM-membrane fractions have been recorded previously and FMM-associated proteins were identified (*García-Fernández et al., 2017*). Revisiting these datasets showed that a high number of iron uptake-associated proteins are enriched in the FMM-containing membrane fraction including the heme transport system IsdEF and the siderophore transporters Fhu, Hts, and Sir.

To investigate further the subcellular localization of IsdF, we generated a *S. aureus* Newman strain that constitutively expresses IsdF with a triple FLAG tag. The cell membrane was isolated and non-ionic detergents were used to collect FMMs, which accumulate in the detergent-resistant membrane (DRM) fraction compared to the detergent-sensitive membrane (DSM) fraction (*Bramkamp and Lopez, 2015*; *Brown, 2002*; *Shah and Sehgal, 2007*). The abundance of proteins in general appeared to be slightly increased in the DSM fraction (*Figure 3A*). In contrast, quantitative western blotting showed

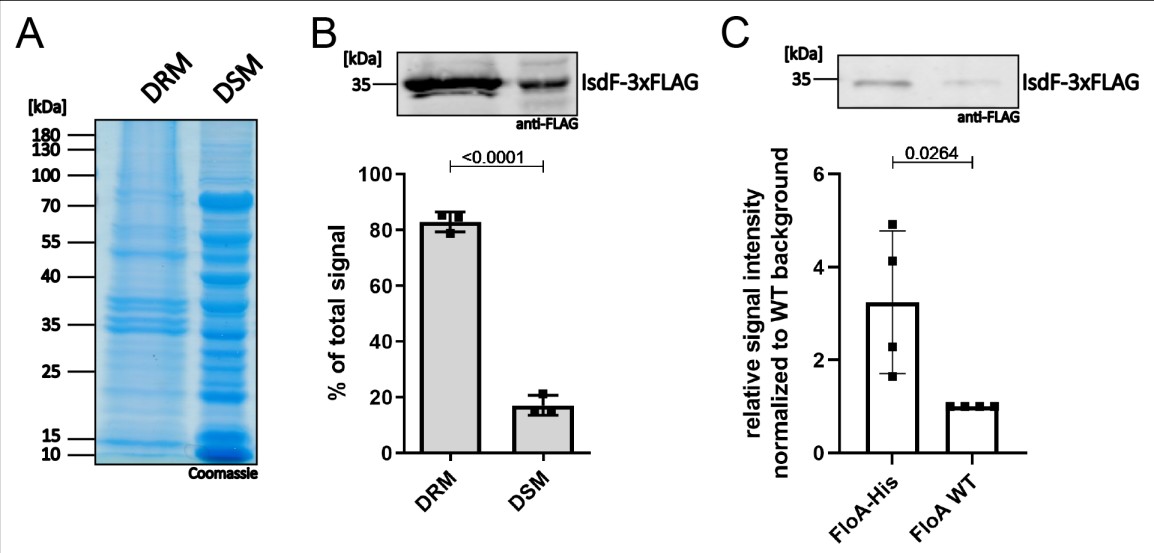

**Figure 3.** IsdF localizes within functional membrane microdomains (FMMs) and interacts directly with flotillin A (FloA). *S. aureus* Newman FloA-His pRB474:*isdF-3xFLAG* membranes were isolated and (**A, B**) separated into detergent-resistant membrane (DRM) and detergent-sensitive membrane (DSM) fractions or (**C**) solubilized with 1% *n*-dodecyl-β-D-maltopyranosid (DDM) overnight and co-precipitated using Ni-NTA affinity chromatography. (**A**) Coomassie blue-stained gel of DRM and DSM fractions (equal amounts loaded). (**B**) Immunoblot analysis and quantification of IsdF in DRM and DSM fractions using anti-FLAG antibody and LI-COR infrared technology. An example of the IsdF-3xFLAG bands in the DRM and DSM fractions is shown. Quantification of signals were calculated as a percentage of the total signal (DRM+DSM). Means and SD of three independent experiments are shown. (**C**) Co-precipitation analysis using anti-FLAG antibody for detection of IsdF-3xFLAG (pRB474:*isdF-3xFLAG*) that co-eluted with FloA-His or FloA WT. An example of the IsdF-3xFLAG bands that co-eluted with FloA-His and FloA WT are shown. Quantification of FloA-His IsdF-3xFLAG signals in immunoblots was normalized to FloA WT strain signals (set to 1). Means and SD of four independent experiments are shown. Statistical analysis (Student's unpaired t-test) was performed using GraphPad Prism 8.

The online version of this article includes the following source data and figure supplement(s) for figure 3:

**Source data 1.** IsdF localizes within functional membrane microdomains (FMMs) and interacts directly with flotillin A (FloA).

**Figure supplement 1.** Input control for co-immunoprecipitation of flotillin A (FloA) and IsdF.

**Figure supplement 1—source data 1.** Input control fro co-immunoprecipitation of flotillin A (FloA) and IsdF.

that 80% of the IsdF signal was detected in the DRM fraction (*Figure 3B*). This confirms that IsdF is predominantly associated with the DRM fraction and suggests that IsdF is localized within FMMs.

It was previously postulated that the structuring protein FloA recruits target proteins to FMMs (*Bramkamp and Lopez, 2015*; *Lopez and Koch, 2017*). Therefore, we investigated if this is also true for IsdF. We co-expressed IsdF-triple FLAG with the FloA WT protein or with FloA tagged with hexa-histidine in *S. aureus* Newman and performed Ni-NTA affinity chromatography. Prior to Ni-NTA chromatography, IsdF was equally abundant in both strains showing equal expression (*Figure 3—figure supplement 1*). However, IsdF was three times more abundant in the Ni-NTA eluate of the FloA-His expressing strain (*Figure 3C*). This strongly suggests that IsdF interacts with FloA which allows co-purification. Low levels of IsdF-triple FLAG were detected within the eluate of the FloA WT expressing strain, suggesting minor non-specific interaction of either IsdF or FloA with the Ni-NTA column.

## FloA and IsdF are spatially coordinated in intact cells

We used fluorescent microscopy to investigate the localization of FloA and IsdF in intact cells of *S. aureus* Newman. The IsdF protein was tagged by mNeongreen while FloA was coupled to SNAP. The latter was visualized by addition of TMR. As described previously, FloA signals formed distinct foci at the bacterial surface most likely representing FMMs (*Figure 4A*). Interestingly, IsdF also accumulated in patches (*Figure 4A*). However, visual inspection indicated that the IsdF and FloA signals did not necessarily overlap perfectly but rather seemed to be in close proximity. We assessed this observation systematically by measuring the distance between proximal fluorescent maxima of IsdF and FloA in hundreds of individual cells (*Figure 4B*). This showed a perfect overlap in 7.5% of measurements while maxima were separated by a single pixel (0.0645 µM) or by two pixels (0.129 µM) in 29.5% and 26.7%

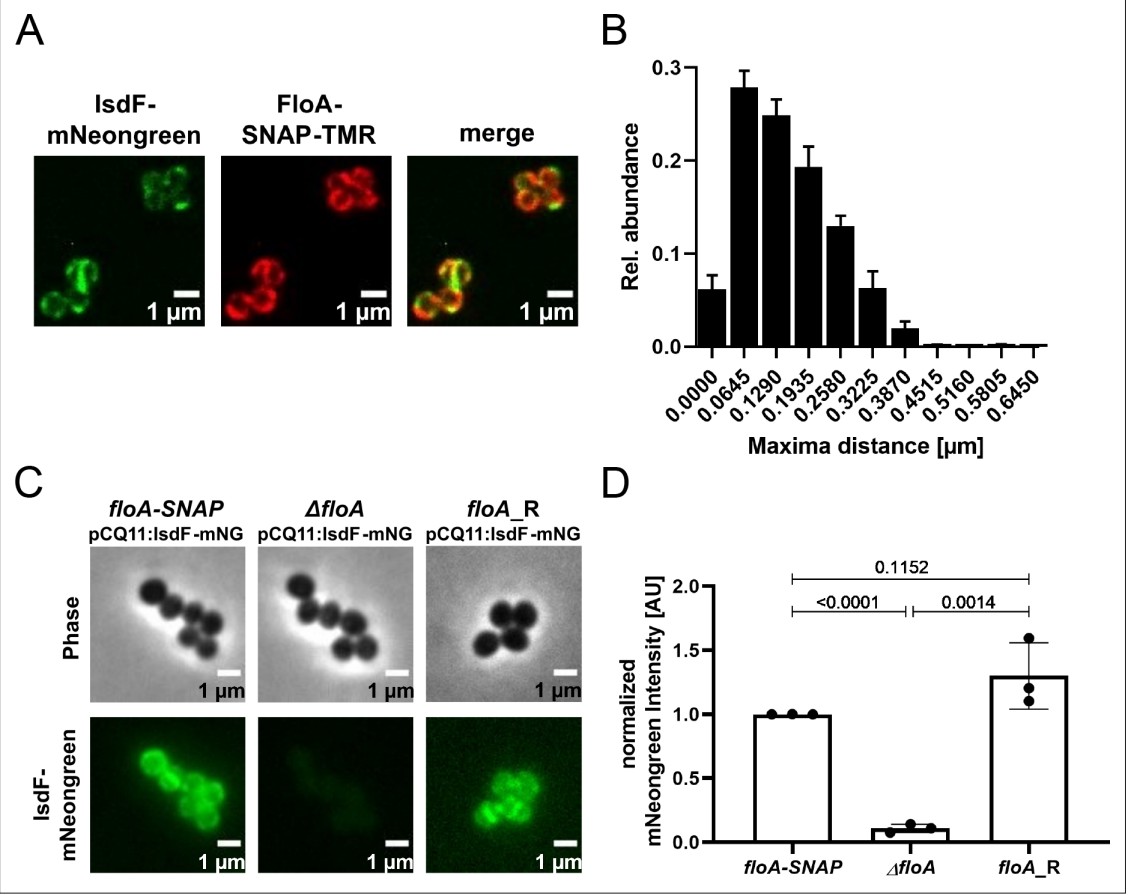

**Figure 4.** Flotillin A (FloA) is crucial for spatial organization of IsdF in the membrane of *S. aureus*. (**A**) Examples of a fluorescence micrograph of *S. aureus* Newman *floA-SNAP* pCQ11:*isdF-mNeongreen*. Green: IsdF-mNeongreen. Red: FloA-SNAP-TMR. Scale bars, 1 μm. (**B**) Quantification of the proximity of IsdF-mNeongreen fluorescence maxima and FloA-SNAP-TMR fluorescence maxima. The distance of each FloA-SNAP-TMR maximum to the nearest IsdF-mNeongreen maximum was measured with pixel (px) accuracy (1 px=0.0645 μm). The histogram shows the relative distribution of determined distances. Bars show means and SD of three independent biological replicates. Total number of maxima measured was n≥876 per replicate for each labeled protein. n≥293 cells per replicate. (**C**) An example of a fluorescence micrograph of *S. aureus* Newman *floA-SNAP* pCQ11:*isdF-mNeongreen*, *ΔfloA* pCQ11:*isdF-mNeongreen*, and *floA*_R pCQ11:*isdF-mNeongreen*. Scale bars, 1 μm. (**D**) Quantification of IsdF-Neongreen fluorescence intensity of individual cells. The bar shows the means and SD of three independent biological experiments. n≥241 cells analyzed per strain. Data was normalized to the respective FloA-SNAP replicate mean. Statistical significance was determined using unpaired two-tailed Student's t-test with 95% confidence interval.

The online version of this article includes the following source data for figure 4:

**Source data 1.** Flotillin A (FloA) is crucial for spatial organization of IsdF in the membrane of *S. aureus.*

of cases, respectively. The number of maxima within a certain distance group declined with increasing distance. Maxima that were further apart than 0.3225 μM were rarely detected. Furthermore, we found that fluorescent signals derived from plasmid pCQ11:*isdF-mNeongreen* were strong in a FloA expressing strain but almost undetectable in a *floA*-deficient mutant (***Figure 4C+D***). Replacement of the mutant allele with the functional WT allele (Newman *floA*_R) by allelic exchange restored IsdF signals to WT level (***Figure 4C+D***). These data suggest that FloA is crucial for the appropriate incorporation and spatial distribution of IsdF within the membrane of *S. aureus*.

## FloA and appropriately formed FMMs are needed for bacterial growth using Hb as the sole iron source

The full functionality of membrane-associated protein complexes in *S. aureus* requires both FMMs and FloA (***García-Fernández et al., 2017***; ***Koch et al., 2017***; ***Mielich-Süss et al., 2017***). Due to the location of IsdF within FMMs, we hypothesized that appropriately formed FMMs would be relevant

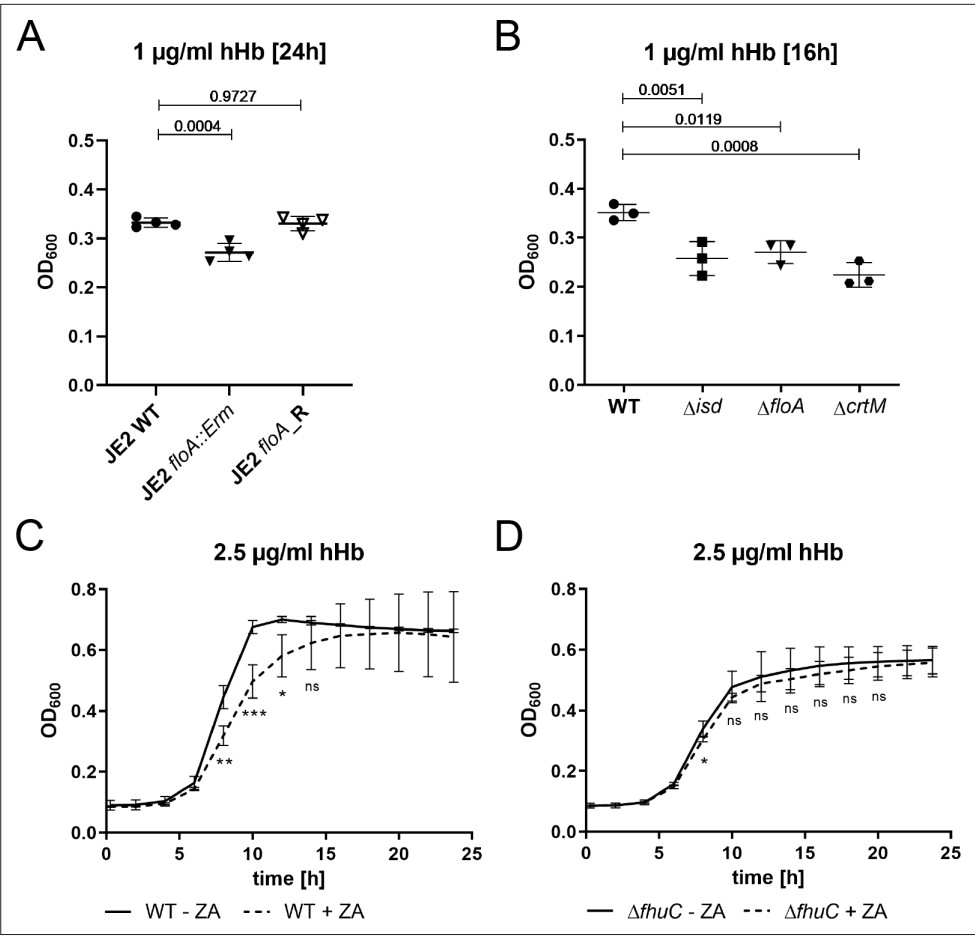

**Figure 5.** Flotillin A (FloA) and functional membrane microdomains (FMMs) are needed for proliferation with hemoglobin. (**A–D**) Strains were grown in iron-limited medium (A: 100 µl in 96-well plates; B–D: 500 µl in 48-well plates). (**A**) Growth of *S. aureus* USA300 JE2 WT, Δ*floA::Erm* and *floA*_Revertant (*floA_R*) in the presence of 1 µg/ml human hemoglobin (hHb). For reasons of clarity, values after 24 hr are displayed. Means and SD of four experiments are shown. (**B**) Growth of *S. aureus* Newman WT, Δ*isd*, Δ*floA*, and Δ*crtM* mutants. Strains were grown in the presence of 1 µg/ml hHb. Values after 16 hr are displayed. Means and SD of three experiments are shown. (**C,D**) Newman WT (**C**) and Δ*fhuC* (**D**) were grown in the presence of 10 µM zaragozic acid (ZA) and 2.5 µg/ml hHb. Values taken every 2 hr are displayed. Means and SD of four experiments are shown. (**A,B**) Statistical analysis: Student's one-way ANOVA followed by Dunett's test for multiple comparisons was performed using GraphPad Prism 9. (**C,D**) Statistical analysis: Student's unpaired t-test was performed using GraphPad Prism 8. *p<0.05, **p<0.01, ***p<0.001.

The online version of this article includes the following source data and figure supplement(s) for figure 5:

**Source data 1.** Flotillin A (FloA) and functional membrane microdomains (FMMs) are needed for proliferation with hemoglobin.

**Figure supplement 1.** FeSO$_4$ growth controls of *floA* and functional membrane microdomain (FMM)-deficient mutants.

**Figure supplement 1—source data 1.** FeSO$_4$ growth controls of *floA* and functional membrane microdomain (FMM)-deficient mutants.

for staphylococcal iron acquisition. First, we tested this using a USA300 JE2 *floA:Erm* mutant derived from the Nebraska transposon mutant library. Growth in the presence of FeSO$_4$ was not impacted by this mutation (**Figure 5—figure supplement 1A**), but a significant growth reduction was observed when Hb was supplied as the sole source of nutrient iron. Replacement of the mutant allele with the functional WT allele (JE2 *floA_R*) by allelic exchange restored full growth (**Figure 5A**). These data support the idea that FloA-dependent incorporation of IsdF is needed for heme acquisition.

Next, we validated the importance of FMMs for heme acquisition also for *S. aureus* Newman. We created three mutant strains lacking (i) the entire Isd system (Δ*isd*), (ii) FloA (Δ*floA*), and (iii) squalene synthase CrtM (Δ*crtM*) which results in FMM deficiency due to the inability to synthesize polyiso-prenoid lipids (*Liu et al., 2012*; *López and Kolter, 2010*). All mutants showed WT levels of growth in the presence of FeSO$_4$ (*Figure 5—figure supplement 1B*) but had a comparable growth deficit in the presence of Hb (*Figure 5B*). Additionally, blockage of FMM biosynthesis by zaragozic acid (ZA) (*López and Kolter, 2010*) reduced Hb-dependent growth (*Figure 5C*). Importantly, the reduced growth of the *fhuC* mutant was not further decreased by the addition of ZA (*Figure 5D*), suggesting that impaired iron acquisition and not any toxic effects of ZA were responsible for the growth defects in the WT strain. In addition, ZA did not affect growth in the presence of FeSO$_4$ (*Figure 5—figure supplement 1C+D*). All these experiments suggest that functional FMMs are crucial for Hb usage by *S. aureus*.

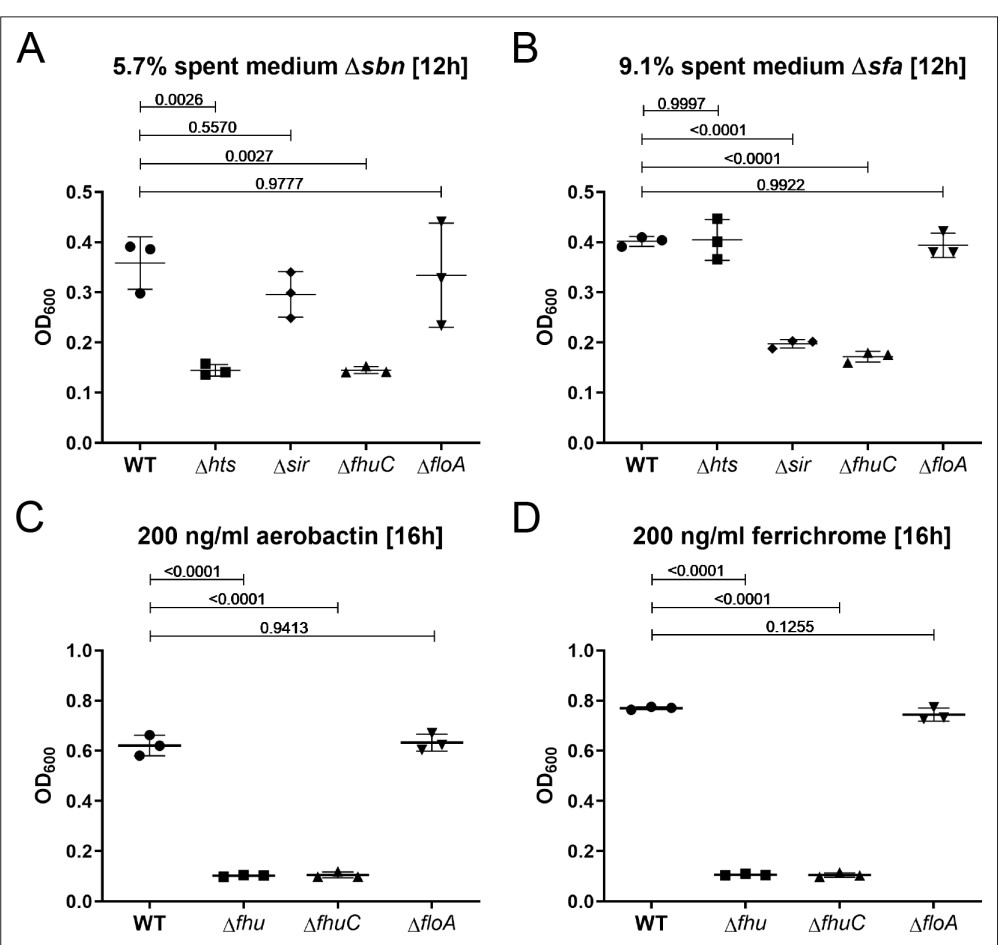

**Figure 6.** Growth using siderophores is independent of functional membrane microdomains (FMMs). Newman WT, Δ*htsABC*, Δ*sirABC*, Δ*fhuCBG*, Δ*fhuC*, and Δ*floA* were grown in the presence of 5.7% spent medium of *S. aureus* USA300 JE2Δ*sbn* (containing staphyloferrin A) (**A**), 9.1% spent medium of USA300 JE2 Δ*sfa* (containing staphyloferrin B) (**B**), 200 ng/ml aerobactin (**C**) or 200 ng/ml ferrichrome (**D**). Strains were grown in 500 µl of iron-limited medium in 48-well plates. For reasons of clarity, values after 12 hr (**A, B**) or 16 hr (**C, D**) are displayed. Means and SD of three experiments are shown. Statistical analysis: Student's one-way ANOVA followed by Dunett's test for multiple comparisons was performed using GraphPad Prism 9.

The online version of this article includes the following source data and figure supplement(s) for figure 6:

**Source data 1.** Growth using siderophores is independent of functional membrane microdomains (FMMs).

**Figure supplement 1.** FeSO$_4$ growth controls of iron transporter and *floA* mutants.

**Figure supplement 1—source data 1.** FeSO$_4$ growth controls of iron transporter and *floA* mutants.

We also found proteins of the siderophore acquisition systems Hts, Sir, and Fhu, to be enriched in the proteomic profile of FMM-membrane fractions (*García-Fernández et al., 2017*). Therefore, we investigated if FloA is also relevant for siderophore acquisition. We created individual mutants of all three systems in *S. aureus* Newman and compared growth of the mutants to that of the *floA* mutant in the presence of SA, SB, or the hydroxamate siderophores aerobactin and ferrichrome. As a source of SA and SB, dilute culture supernatant of *S. aureus* USA300 JE2 Δ*sbn* (secreting only SA) and *S. aureus* USA300 JE2 Δ*sfa* (secreting only SB) was used, respectively. All mutants showed WT levels in the presence of $FeSO_4$ (*Figure 6—figure supplement 1A+B*). The Δ*htsABC* and Δ*sirABC* mutants showed growth deficiency in the presence of SA and SB (*Figure 6A+B*), respectively, and Δ*fhuCBG* failed to grow in the presence of aerobactin and ferrichrome (*Figure 6C+D*). The *fhuC* mutant showed growth deficiency with all siderophores tested (*Figure 6A–D*). These data are in agreement with previous datasets (*Beasley et al., 2009*; *Cheung et al., 2009*; *Sebulsky et al., 2000*; *Speziali et al., 2006*). However, the Δ*floA* mutant did not show any growth deficits compared to the WT strain under the tested conditions (*Figure 6A–D*). This data suggests that in contrast to acquisition of Hb-derived heme, acquisition of siderophores is not dependent on membrane structuring by FMMs.

## Sortase function does not depend on FloA and FMMs

A major difference between the heme membrane transporter IsdEF and all siderophore transport systems is that IsdEF relies on CWAs (IsdA, IsdB, IsdH, IsdC) to extract heme from Hb and to funnel it over the cell wall to the membrane transporter (*Mazmanian et al., 2003*). In contrast, siderophores diffuse freely to the cell membrane. CWAs are anchored to the cell wall by the action of sortases (*Marraffini et al., 2006*). Intriguingly, when we reanalyzed the FMM proteomic datasets of García-Fernández and colleagues, we found both sortases of *S. aureus* (sortase A [SrtA] and sortase B [SrtB]) to be enriched within the FMM fraction (*García-Fernández et al., 2017*). Accordingly, it seemed possible that the Hb-dependent growth defects of FMM and *floA* mutants might in part result from impaired sorting of CWA Isd proteins which could hinder heme extraction and funneling. To investigate this, we assessed the functionality of the housekeeping SrtA by studying the cellular localization of its substrate IsdA. We separated cell wall and membrane fractions of various strains and detected IsdA by western blotting. As reported earlier (*Mazmanian et al., 2003*), IsdA was localized in the cell wall fraction of *S. aureus* WT. In the *srtA* mutant, the protein was exclusively detected in the membrane fraction (*Figure 7*). Interestingly, neither inactivation of FloA nor of CrtM resulted in mislocalization of IsdA (*Figure 7*). These data show that sorting of IsdA is independent of FMMs, suggesting general functionality of the sortases.

## FloA is needed for Isd-dependent Hb usage in *S. lugdunensis*

As Isd systems are found in several staphylococcal species (*Heilbronner et al., 2016*; *Mazmanian et al., 2003*; *Sun et al., 2020*), we investigated if the involvement of FMMs and flotillins in heme uptake via the Isd system is a general concept. *S. lugdunensis* encodes for an Isd system similar to that of *S. aureus* (*Heilbronner et al., 2011*; *Heilbronner et al., 2016*). However, in addition to the Isd system, *S. lugdunensis* encodes the energy coupling factor-type heme transporter LhaSTA

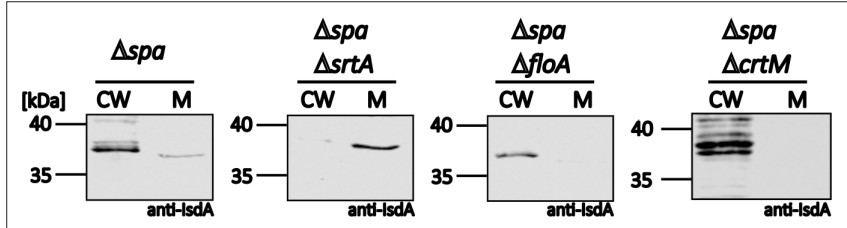

**Figure 7.** Sortase function does not depend on functional membrane microdomains (FMMs). *S. aureus* Newman Δ*spa*, Δ*spa*Δ*srtA::Erm*, Δ*spa*Δ*floA::Erm,* and Δ*spa*Δ*crtM::Erm* cells were grown in iron-limited medium and treated with lysostaphin to gain cell wall (**CW**) and membrane (**M**) fractions. Fractions were analyzed by SDS-PAGE and western blotting. IsdA was detected using polyclonal anti-IsdA antibodies.

The online version of this article includes the following source data for figure 7:

**Source data 1.** Sortase function does not depend on functional membrane microdomains (FMMs).

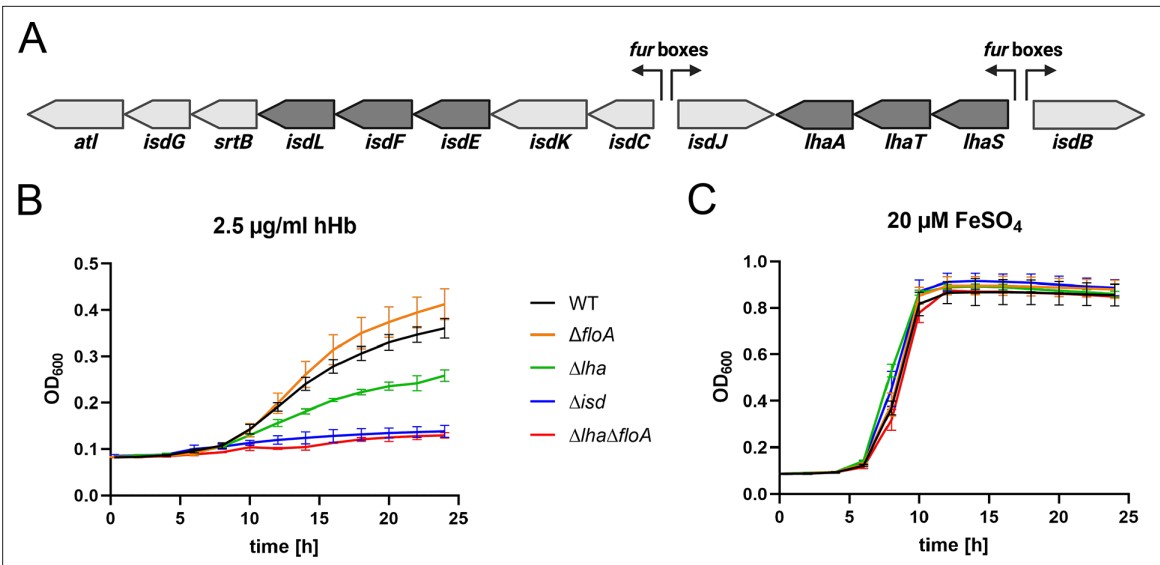

**Figure 8.** Flotillin A (FloA) is needed for iron-regulated surface determinant (Isd)-dependent proliferation in *S. lugdunensis*. (**A**) Schematic representation of the *isd* locus in *S. lugdunensis* N920143. Membrane transporters IsdEFL and LhaSTA in dark gray. Fur boxes are indicated. This figure was created with https://www.biorender.com/. (**B, C**) *S. lugdunensis* N920143 WT, Δ*floA*, Δ*lha*, Δ*isd*, and Δ*lha*Δ*floA* were grown in 500 µl of iron-limited medium in the presence of 2.5 µg/ml human hemoglobin (hHb) (**B**) or 20 µM $FeSO_4$ (**C**) as the sole source of iron in 48-well plates. Values taken every 2 hr are displayed. Means and SD of three experiments are shown.

The online version of this article includes the following source data for figure 8:

**Source data 1.** Flotillin A (FloA) is needed for iron-regulated surface determinant (Isd)-dependent proliferation in *S. lugdunensis*.

(*Jochim et al., 2020*; *Figure 8A*). Inactivation of *lhaSTA* in *S. lugduneneis* N920143 resulted in a significant reduction of Hb-dependent growth but only deletion of the entire *isd* locus (*atl* to *isdB* including *lhaSTA*) abrogated growth completely (*Figure 8B*). Accordingly, growth of the Δ*lhaSTA* mutant reflects Isd-dependent proliferation in *S. lugdunensis*. We identified a single FloA homologue (SLUG_13380) in *S. lugdunensis*. Inactivation of *floA* in the Δ*lhaSTA* background resulted in complete abrogation of Hb-dependent growth (*Figure 8B*), showing the importance of functional FMMs for Isd-dependent heme in this species, too. Interestingly, inactivation of *floA* in *S. lugdunensis* WT did not influence Hb-dependent growth, suggesting that activity of LhaSTA does not depend on FMMs and allows inhibition of the Isd system to be by-passed. All mutants showed WT levels of growth in presence of $FeSO_4$ (*Figure 8C*).

## Discussion

ABC transporters are classical molecular machines that transport nutrients across biological membranes and allow their accumulation against concentration gradients at the cost of ATP (*Rees et al., 2009*). Prokaryotic ABC-type importers consist of three functionally distinct subunits, an extracellular substrate-binding protein (SBP), a membrane-located permease and a cytosolic ATPase. In Gram-negative bacteria, SBPs are located in the periplasmic space while the SBPs of Gram-positive bacteria are lipoproteins that are coupled to the extracellular leaflet of the membrane. SBPs bind the target substrate and deliver it to the membrane. The membrane permeases consist of two subunits (homo- or heterodimers) and promote translocation of the substrate. The ATP-binding protein binds to the permease and energizes transport of the substrate by hydrolysis of ATP (*Dassa and Bouige, 2001*; *Locher, 2009*; *Zolnerciks et al., 2011*). It is known that a single ATPase can energize several permeases with different substrates (*Quentin et al., 1999*) including the iron compound permeases of *S. aureus*. *S. aureus* possesses genes for biosynthesis of the siderophores SA and SB (*Beasley et al., 2009*; *Cheung et al., 2009*; *Cotton et al., 2009*). However, the loci encoding their respective importers do not express an ATPase. It has been shown that FhuC, the ATPase encoded within the hydroxamate siderophore transport system *fhuCBG*, is needed for staphyloferrin-dependent proliferation (*Beasley et al., 2009*; *Sebulsky et al., 2000*; *Speziali et al., 2006*). The *isd* operon of *S. aureus* also lacks a

gene encoding an ATPase. Here, we show that deletion of *fhuC* has a major impact on Hb-dependent growth. This suggests that FhuC is a housekeeping ATPase that powers acquisition of different iron compounds by *S. aureus*. However, weak Hb-dependent growth was observed with the Δ*fhuC* mutant. This could indicate that an unknown ATPase partially substitutes for the function of FhuC. Such an exchangeability of ATPases was previously described (*Hekstra and Tommassen, 1993*; *Leisico et al., 2020*; *Webb et al., 2008*). Seven ABC iron compound transporters were previously identified in the genome of *S. aureus* (Sst, Fhu, Sir, Hts, Isd systems, SAUSA300_1514–1517 and SAUSA300_0598–99) with SstC and SAUSA300_1516 being ATPases besides FhuC (*Skaar et al., 2004*). It seems possible that these are able to partly substitute for FhuC. However, further experiments are needed to validate this. Alternatively, a secondary heme transporter might exist in *S. aureus*. It is worth considering that Isd-dependent heme acquisition in other pathogens appears to be energized by specialized ATPases. This has been experimentally proven for *S. lugdunensis* where the ATPase IsdL (*Heilbronner et al., 2011*) energizes heme transport by IsdEF (*Flannagan et al., 2022*). Similarly, the *isd* loci of *B. anthracis* (*Skaar et al., 2006*), *B. cereus* (*Abi-Khalil et al., 2015*), and *L. monocytogenes* (*Jin et al., 2006*; *Klebba et al., 2012*) possess genes encoding putative ATPases. However, the relevance of these ATPases for heme acquisition lacks experimental validation. Nevertheless, it seems plausible that they are required to energize heme membrane transport in these organisms.

All available knowledge indicates that FhuC functions as a housekeeping ATPase that energizes at least four different iron compound transporters (FhuCBGD$_1$D$_2$, SirABC, HtsABC, and IsdEF). Similar cases of ATPases energizing a variety of importers have been reported for other species such as *Streptococcus pneumoniae* (*Linke et al., 2013*; *Marion et al., 2011*), *Streptococcus suis* (*Tan et al., 2015*), *Streptomyces reticuli* (*Schlösser, 2000*), *Streptomyces lividans* (*Hurtubise et al., 1995*), and *Bacillus subtilis* (*Ferreira et al., 2017*; *Ferreira and Sá-Nogueira, 2010*; *Morabbi Heravi et al., 2019*; *Schönert et al., 2006*). All these systems are carbohydrate type I importers while those energized by FhuC are type II importers. Type I importers typically have permeases with only six transmembrane domains (TMDs) and acquire substrates like ions, amino acids, and sugars. In contrast, type II importer permeases consist of 10 TMDs and take up bigger substrates like heme or cobalamin (*Locher, 2009*). However, precise structural motifs that allow interactions between an ATPase and multiple different permeases remain widely elusive. ATPases and permeases interact using Q-loops and coupling helices, respectively (*Hollenstein et al., 2007*; *Locher, 2009*). Coupling helices often contain an EAA motif to facilitate interaction with an ATPase (*ter Beek et al., 2014*). A classical EAA motif is not apparent within the coupling helices of FhuB, FhuG, HtsB, HtsC, SirB, SirC, and IsdF. However, we identified conserved alanine and glycine residues that in combination are necessary for interaction with FhuC, which suggests that these residues are of crucial importance for targeting FhuC. Besides conserved amino acids, the secondary structures of coupling helices are important for the recruitment of ATPases (*Beis, 2015*; *Hollenstein et al., 2007*; *Locher, 2009*). However, the full-length coupling helix of IsdF, when inserted into MntB, did not allow MntB to interact with FhuC. This suggests that additional molecular signatures enable FhuC to identify appropriate interaction partners. Interestingly, deletion of the C-terminal part of IsdF including the fourth cytosolic loop prevented interaction with FhuC. It is possible that truncation of the C-terminal part of IsdF prevents appropriate folding or homodimerization, which might inhibit appropriate interaction with the ATPase. Nevertheless, it is also possible that in addition to a matching coupling helix, interaction between C-terminal residues of IsdF and FhuC is needed to stabilize binding. Further experiments to investigate the molecular structure of the protein complexes by X-ray crystallography are needed to validate this hypothesis.

Interestingly, components of all FhuC-energized transporters are associated with the DRM fraction of *S. aureus* membranes, suggesting that they are integrated in FMMs. FMMs were previously shown to be important for the oligomerization of cell wall biosynthetic enzymes like PBP2a (*García-Fernández et al., 2017*), for the function of membrane-bound protein complexes like the type VII secretion system (*Mielich-Süss et al., 2017*) and the RNase Rny, which is part of the degradosome (*Koch et al., 2017*) in *S. aureus*. A role for FMMs in nutrient acquisition has not been described previously. Many different ABC transporters are present in the proteomic profile of FMM-membrane fractions, for example components of magnesium (MgtE), manganese (MntH), molybdenum (ModA), phosphate (PitA), oligopeptide (OppB, OppC), and fructose (FruA) transporters (*García-Fernández et al., 2017*). It is unclear if this is the result of biochemical characteristics that are shared between membrane proteins or if this association is promoted by the FMM scaffolding protein FloA and is

important for import of nutrients. Here, we showed that FloA directly interacts with the permease IsdF, supporting the idea of active integration of permeases into FMMs. If this holds true for other permeases remains to be investigated. However, in the light of shared ATPases it is tempting to speculate that active integration into FMMs can create spatial proximity between permeases and improve the recruitment of energizing proteins such as FhuC to optimize functionality. It is possible that FloA facilitates oligomerization of larger complexes consisting of multiple permeases that are collectively energized. However, we found that disruption of FMMs by mutation of *floA* interfered with Isd-dependent proliferation but not siderophore-dependent growth. Similarly, we found that the function of sortase is not impaired in a *floA*-deficient mutant. This indicates that active integration into FMMs is not occurring for all proteins identified by DRM/DSM analysis in membrane extracts. But it is clearly occurring and functionally relevant for some of them.

Previously, treatment with ZA was shown to reduce *S. aureus* virulence in vitro and in vivo (*García-Fernández et al., 2017*; *Koch et al., 2017*; *Mielich-Süss et al., 2017*). Our data suggest that besides reducing the functionality of virulence factors, limiting the ability to overcome restriction of iron acquisition might also contribute to this phenomenon. Accordingly, statins like ZA might represent a new treatment approach to treat *S. aureus* infections by targeting diverse cellular functions simultaneously

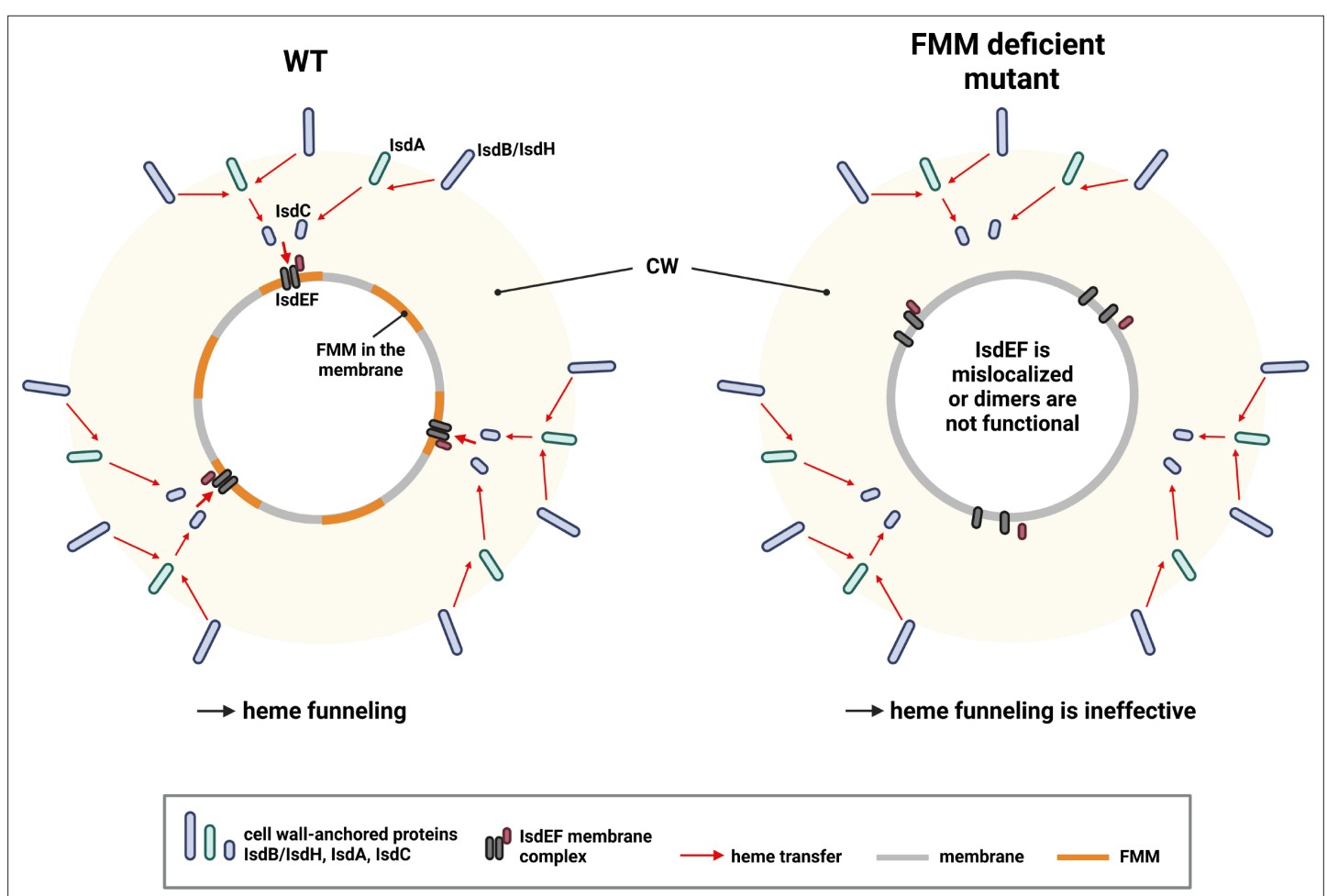

**Figure 9.** Proposed model of heme funneling over the *S. aureus* cell envelope. Cell wall (CW) and membrane of a *S. aureus* are shown. Heme transfer is indicated by red arrows. The surface-exposed receptors (IsdB and IsdH) extract heme from host hemoproteins and guide it to IsdA and IsdC. We propose that functional membrane microdomains (FMMs) allow structural alignment of IsdC and the membrane receptor IsdEF. Alternatively, the IsdEF complex might be unstable in an FMM or flotillin A (FloA)-deficient strain. This figure was created with https://www.biorender.com/.

The online version of this article includes the following source data for figure 9:

**Source data 1.** Proposed model of heme funneling over the *S. aureus* cell envelope.

and could lead to the resurrection on β-lactam antibiotics to treat MRSA infections (*Foster, 2019*; *Somani et al., 2016*).

## Speculation

It is unclear why activity of Isd depends on FMMs while siderophore transport does not. However, there must be a functional difference between the systems explaining this. Only Isd-dependent heme acquisition depends on the activity of CWAs. Interestingly, the Isd-dependent heme transport across the cell wall is structured. Hb binding and heme extraction is facilitated by the surface-exposed receptors IsdB and IsdH. Heme is then passed to IsdA and from there to IsdC. In contrast to the other cell wall-anchored molecules, IsdC is anchored by SrtB, allowing placement of the protein in the central layers of the peptidoglycan (*Mazmanian et al., 2003*; *Mazmanian et al., 2002*). This structure is referred to as a 'heme funnel' (*Sheldon and Heinrichs, 2015*) and is thought to allow concerted passage of heme over the envelope. It seems possible that the funnel needs the heme-specific transporter IsdEF to be appropriately aligned with IsdC to allow efficient passage of heme to the membrane (*Figure 9*). Along this line, FMMs might allow alignment of IsdC and IsdEF. However, a deeper understanding of the spatial location and distribution of IsdC as well as of FMMs is needed to verify this.

Alternatively, it seems possible that FloA is crucial for formation of the IsdF homodimer within the membrane. It was previously shown that FloA deficiency reduces the formation of protein complexes in the membrane of *S. aureus* (*García-Fernández et al., 2017*; *Koch et al., 2017*; *Mielich-Süss et al., 2017*). Our microscopy analysis showed that IsdF-derived signals were almost indetectable within a *floA*-deficient background, which might result from an incomplete folding or homodimer formation which might entail degradation of the protein.

## Materials and methods

### Chemicals

If not stated otherwise, reagents were purchased from Sigma.

### Bacterial strains, media, and culture conditions

The bacterial strains generated and used in this study are listed in Appendix—Key resources table. *E. coli* strains were grown in Lysogeny Broth (LB), *S. aureus* strains in TSB or RPMI+1% casamino acids (CA) (Bacto, BD Biosciences) overnight at 37°C with agitation. Antibiotics were added where appropriate: kanamycin (50 μg/ml), ampicillin (100 μg/ml), chloramphenicol (10 μg/ml), erythromycin (2.5 μg/ml).

### Creation of deletion mutants, complementation, 6xHis-/3xFLAG-/SNAP-/mNeongreen-tagged strains

To create markerless deletions of *S. aureus* Newman *isd*, *fhu (fhuCBG)*, *hts (htsABC)*, *sir (sirABC)*, *fhuC*, *floA*, and *spa*, and of *S. aureus* USA300 JE2 Δ*sbn (sbnA-I)* and Δ*sfa (sfaA-D)*, 500 bp DNA flanking regions of the genes to be deleted were amplified from chromosomal DNA. For the genomic complementation of USA300 JE2 *floA::Erm* (*floA_R*) and Newman Δ*floA* (*floA_R*), 500 bp upstream with the first half of the *floA* gene and the second half of the *floA* gene and 500 bp downstream were amplified. PCR fragments were fused by overlap extension PCR and cloned into pIMAY by restriction digestion.

For the deletion of *floA* in *S. lugdunensis* N920143, we identified the *S. aureus* Newman *floA* homologue (SLUG_13380) in *S. lugdunensis* N920143 using BLAST. Five-hundred bp upstream and downstream of the gene were amplified, joint by overlap extension PCR and cloned into pIMAY.

For the creation of the C-terminally 6xHis-tagged *floA-6xHis* and SNAP-tagged *floA-SNAP* strains, the 3 500 bp of the *floA* gene as well as the 500 bp of the downstream region of *floA* stop codon were amplified by PCR. For the 6xHis-tagged strain, the primers contained a sequence overlap and in addition a linker (AGAGGATCG) and the hexa histidine encoding sequence (CATCACCATCACCATCAC) to integrate the tag before the stop codon of the *floA* gene; for the SNAP-tagged strain, the primers contained a sequence overlap for the SNAP sequence, which was amplified from pCQ11:*snap*, to integrate the tag before the stop codon of *floA*. Fragments were joint by overlap extension PCR and cloned into pIMAY using restriction digestion.

Allelic replacement was used to create staphylococcal mutants as described elsewhere (*Monk et al., 2012*).

For the complementation of Newman Δ*fhuC*, the gene including its fur box was amplified and cloned into pRB473 using restriction digestion. The final plasmid was used to transform *S. aureus* Newman using standard procedure.

For the expression of IsdF-3xFLAG, a fragment encompassing *isdF* and its ribosomal binding site was amplified. The 3xFLAG encoding sequence was amplified from pRB474:*mprFdelCysflag*; both fragments were combined by overlap extension PCR and cloned into pRB474 using restriction digestion. The final plasmid was used to transform *S. aureus* Newman.

For the expression of IsdF-mNeongreen, first pCQ11:*mNeongreen* was constructed by restriction digest substitution of *gfp* in pCQ11:*gfp* with *mNeongreen* obtained from pLOM-S-mNeongreen-EC18153 (Julian Hibberd, Addgene plasmid # 137075; http://n2t.net/addgene:137075; RRID: Addgene_137075). pCQ11:*isdF-mNeongreen* was constructed via restriction digest, *isdF* was obtained via PCR from *S. aureus* COL genomic DNA. pCQ11:*isdF-mNeongreen* was transformed into *E. coli* SA08B as shuttle strain for further cloning into *S. aureus* Newman strains.

Oligonucleotides and endonuclease restriction sites are shown in Appendix–Key resources table.

## Mutagenesis using phage transduction

*S. aureus* Newman Δ*spasrtA::Erm*, Δ*spafloA::Erm*, and Δ*spacrtM::Erm* mutants were created using phage transduction (phage Φ11). Transductions of the respective mutations from the Nebraska transposon mutant library into the markerless deletion mutant Newman Δ*spa* were performed according to the standard transduction protocols.

## Plasmid construction for BACTH and β-galactosidase assay

*S. aureus* USA300 LAC WT chromosomal DNA was used as template to amplify *fhuB*, *mntB*, *isdF*, and *fhuC*. The fragments were cloned into the pKT25 and pUT18C vectors (Euromedex) by restriction digestion. A nonsense codon was integrated in the pKT25:*mntB* construct to terminate translation.

To create truncations of IsdF, the topology of IsdF was predicted using the online tool TOPCONS (*Tsirigos et al., 2015*). pKT25:*isdF* was used as a template and primers to truncate the protein after aa131, aa203, aa260, and aa313 were designed. A KpnI restriction site was incorporated into each primer to allow religation of the plasmid after PCR amplification.

For the exchange of alanine position 213 and glycine position 217 to phenylalanines (*isdF_A213F*, *isdF_G217F*, *isdF_A+G_F*), pKT25:*isdF* was used as PCR template with primers including respective point mutations for the site-directed mutagenesis. The PCR product was digested using DpnI for 3 hr at 37°C, and subsequently, *E. coli* XL-1 blue was transformed with the obtained plasmid.

For the exchange of the coupling helix of MntB to the one from IsdF, *mntB* was amplified from the template pKT25:*mntB* using primers that contained the coupling helix sequence of *isdF* instead of the original sequence via overlap extension PCR. The obtained *mntB_CH$_{isdF}$* was cloned into the original pKT25:*mntB* plasmid via restriction digest after excising *mntB* from it.

*E. coli* XL-1 blue was transformed with the different plasmids and the sequence was confirmed by Sanger sequencing.

Oligonucleotides and endonuclease restriction sites are shown in Appendix–Key resources table.

## Topology prediction, alignments, and visualization

The topologies of the permeases were predicted using TOPCONS (*Tsirigos et al., 2015*). Additionally, the structure of IsdF was predicted using Alphafold (*Jumper et al., 2021*; *Varadi et al., 2022*) and visualized using PyMOL (The PyMOL Molecular Graphics System, Version 2.5, Schrödinger, LLC). Coupling helixes of permeases were aligned using Clustal Omega.

## Purification of hHb

hHb was purified as described elsewhere (*Pishchany et al., 2013*).

## Spent media containing SA and SB

SA- and SB-containing spent media were obtained from *S. aureus* USA300 JE2 Δ*sbn* (deletion of SB biosynthesis genes) and Δ*sfa* (deletion of SA biosynthesis genes), respectively. Strains were grown in

BHI with 10 µM of the iron chelator ethylenediamine-N,N′-bis(2-hydroxyphenylacetic acid) (EDDHA) (LGC Standards) to induce expression of iron-regulated genes for 3 days at 37°C with agitation. The supernatants were collected, sterile filtered, and the obtained spent media were used as SA- or SB-iron source in growth assays.

## Growth in iron-limited medium

Staphylococcal deletion mutant strains were grown overnight in TSB at 37°C with agitation. Cells were harvested and washed with RPMI containing 1% CA and 10 µM EDDHA. $OD_{600}$ was adjusted to 1 and 2.5 µl were mixed with 500 µl of RPMI+1% CA+10 µM EDDHA per well (final $OD_{600}$ of 0.005) in a 48-well microtiter plate (Nunc, Thermo Scientific) or 0.5 µl were mixed with 100 µl RPMI+1% CA+10 µM EDDHA per well in a 96-well microtiter plate (Falcon flat bottom, Fisher Scientific), respectively. One µg/ml, 2.5 µg/ml hHb (own purification), 200 ng/ml aerobactin (EMC Microcollections), 200 ng/ml ferrichrome (EMC Microcollections), 5.7% spent medium from USA300 JE2 Δsbn containing SA, 9.1% spent medium from USA300 JE2 Δsfa containing SB, or 20 µM $FeSO_4$ were added as iron sources. 10 µM ZA (Santa Cruz Biotechnology) were used to inhibit membrane microdomain assembly (stock dissolved in 1:1 DMSO:methanol) or as a control the same amount of DMSO/methanol. $OD_{600}$ was measured every 15 min for 24 hr in an Epoch2 reader (BioTek) at 37°C orbital shaking.

## BACTH assay

To investigate interactions between the ATPase FhuC and different permeases, the commercially available BACTH kit was used (Euromedex). In brief, *E. coli* BTH101 was co-transformed with the plasmid pKT25 expressing one of the permeases and pUT18C:*fhuC*. The vectors encode for the T25 and T18 catalytic domain of *Bordetella pertussis* adenylate cyclase, respectively. In case of direct interaction of the proteins of interest, these catalytic domains heterodimerize producing cyclic AMP (cAMP) leading to *lacZ* expression. This can be detected as blue colony formation on indicator plates consisting of LB agar 40 µg/ml X-Gal, 0.5 mM isopropyl β-D-1-thiogalactopyranoside (IPTG) (Thermo Scientific), 100 µg/ml ampicillin, and 50 µg/ml kanamycin after 1 day at 30°C. As negative controls, empty vectors were used. As positive control, pKT25:*zip*+pUT18C:*zip* were used encoding a leucine zipper.

## β-Galactosidase assay

β-Galactosidase activity was measured to quantify the protein-protein interaction seen in BACTH assay. This activity correlates with the production of cAMP by heterodimerization of T25 and T18 domains. The measurement was performed similarly as described previously (*Griffith and Wolf, 2002*). In brief, *E. coli* BTH101 co-transformed with pKT25 and pUT18C plasmids were inoculated in 800 µl of LB containing 0.5 mM IPTG, 100 µg/ml ampicillin, and 50 µg/ml kanamycin overnight at 37°C with agitation. $OD_{600}$ was measured, and 200 µl of the overnight cultures were mixed with 1 ml of buffer Z (60 mM $Na_2HPO_4$, 40 mM $NaH_2PO_4$, 10 mM KCl, 1 mM $MgSO_4$, 50 mM β-mercaptoethanol), 40 µl of 0.1% SDS, and 80 µl of chloroform for 30 min at room temperature to allow permeabilization of cells. One-hundred µl of the aqueous upper phase were transferred to a 96-well microtiter plate, and 20 µl of 4 mg/ml 2-nitrophenyl β-D-galactopyranoside (ONPG) were added to start the calorimetric reaction. $OD_{420}$ and $OD_{550}$ were measured for 4 hr. The highest $OD_{420}$ and its corresponding $OD_{550}$ and t values were used. Blanks were substracted from $OD_{600}$ (blank = LB), $OD_{420}$ and $OD_{550}$ (blank = 100 µl of buffer Z, 0.1% SDS, chloroform, ONPG). The β-galactosidase activity was calculated using the formula $1000*((OD_{420} - 1.75*OD_{550})/(t*v*OD_{600}))$ with t=reaction time in min and v=reaction volume 0.12 ml.

## Isolation of cell membranes of *S. aureus*

Strains were grown overnight in RPMI+1% CA at 37°C with agitation. Cells were harvested, resuspendend in 10 ml of PBS buffer with 5 µg/ml DNaseI (from bovine pancreas, Roche) and 10 µg/ml lysostaphin, and incubated for 20 min at 37°C to allow cell lysis. One mM phenylmethylsulfonylfluoride (Roth) was added, and cells were disrupted by adding 5 ml of glass beads using a FastPrep-24 (MP Biomedicals) for two times 40 s 6.5 m/s. Unbroken cells and debris were removed by centrifugation for 10 min 11,000 × g at 4°C. The supernatant was ultracentrifuged for 1 hr 100,000 × g at 4°C to separate the membrane fraction. The pelleted membrane fraction was dissolved overnight at 4°C for further analysis.

## DRM/DSM assay

For separation of cell membranes into DRM and DSM, the CelLytic MEM protein extraction kit (Sigma) was used as described elsewhere (*López and Kolter, 2010*). Cell membranes were isolated as described above and dissolved overnight rotating at 4°C in 600 µl lysis and separation working solution. Equal amounts of DRM and DSM fractions were analyzed by SDS-PAGE followed by Coomassie staining and western blotting for detection of IsdF-3xFLAG using an anti-FLAG M2 antibody (Sigma, #F3165) and infrared imaging (Odyssey CLx, LI-COR). Using Image Studio, IsdF-3xFLAG signals were quantified; with Microsoft Excel and GraphPad Prism 8, DRM and DSM signal intensities were calculated as amount of the total signal (DRM+DSM).

## Co-immunoprecipitation

During the co-immunoprecipitation assays, samples were kept at 4°C throughout the experiment. After isolation of cell membranes, 12 mg were dissolved overnight rotating at 4°C in 2 ml of 50 mM Tris HCl pH 8, 250 mM NaCl, 1% *n*-dodecyl-β-D-maltopyranosid (DDM) (Roth). Unsolubilized membrane was removed by centrifugation at 16,200 × *g* 20 min at 4°C. The supernatant was added to a polypropylene column (Bio-Rad) containing 500 µl of profinity IMAC resin Ni-charged (Bio-Rad) and mixed for 30 min rotating at 4°C. The Ni resin slurry was equilibrated with 10 column volumes (CV) of 50 mM Tris HCl pH 8, 250 mM NaCl before. The column was washed with 10 CV of 50 mM Tris HCl pH 8, 250 mM NaCl, 0.04% DDM, and 10 CV of the same buffer with 10 mM imidazole. Bound proteins were eluted with 1 ml of 50 mM Tris HCl pH 8, 250 mM NaCl, 0.04% DMM, 50 mM imidazole. The elution fraction was precipitated with 10% trichloroacetic acid (Merck) and analyzed by western blotting for detection of 3xFLAG-tagged IsdF using an anti-FLAG M2 antibody (Sigma, #F3165) and infrared imaging (Odyssey CLx, LI-COR). Using Image Studio, IsdF-3xFLAG signals were quantified. Using Microsoft Excel and GraphPad Prism 8, IsdF-3xFLAG signals were calculated as ratio between the FloA-His strain and control strain with the control strain set to 1.

## Cell fractionation

For separation of bacterial cells into cell wall, membrane, and cytosolic fraction, *S. aureus* Newman strains were grown in RPMI+1% CA overnight. Fractionation was performed as described elsewhere (*Heilbronner et al., 2016*). Cell wall and membrane fractions were analyzed by SDS-PAGE followed by western blotting. IsdA was detected using rabbit serum anti-IsdA and infrared imaging (Odyssey CLx, LI-COR).

## Fluorescence microscopy

For fluorescence microscopy of IsdF-mNeongreen and FloA-SNAP, main culture of *S. aureus* Newman *floA-SNAP* pCQ11:*isdF-mNeongreen* (or Newman Δ*floA* pCQ11:*isdF-mNeongreen* and Newman *floA*_R pCQ11:*isdF-mNeongreen* as controls) was inoculated to $OD_{600}$=0.05 from an overnight culture and grown in MH medium at 37°C under constant shaking for 6 hr. Induction of *isdF-mNeongreen* expression was achieved with 0.1 mM IPTG. FloA-SNAP was labeled with 0.2 µM SNAP-Cell TMR-Star (New England Biolabs, #S9105S) during the last 20 min of main culture incubation. All samples were washed three times in MH medium prior to microscopy. Cells were transferred to a microscopy slide coated with 1% agarose for microscopy. Microscopy was performed on a Carl Zeiss AxioObserver Z1 equipped with a X-Cite Xylis LED Lamp, an αPlan-APOCHROMAT ×100/1.46 oil immersion objective and an AxioCam MRm camera. Visualization IsdF-mNeongreen was achieved using Carl Zeiss filter set 38 (450–490 nm excitation, 495 nm beam splitter, and 500–500 nm emission). Visualization of SNAP-Cell TMR-Star was achieved using Carl Zeiss filter set 43 (538–562 nm excitation, 570 nm beam splitter, and 570–640 nm emission). Deconvolution was achieved using Carl Zeiss Zen 3.6 direct processing. Image analysis was performed using Fiji (ImageJ) Version 2.0.0-rc-69/1.52p; Java 1.8.0_172 (64-bit) (*Schindelin et al., 2012*) and the ImageJ plug-in MicrobeJ 5.13n (9) – beta (*Ducret et al., 2016*). Statistical analysis was performed with GraphPad Prism 8.0.2 (263).

## Acknowledgements

The authors thank Prof. J Geoghegan for providing anti-IsdA antiserum and Gina Marke for construction of plasmid pCQ11-IsdF-mNeongreen. The authors thank Libera Lo Presti and Timothy J Foster

for critically reading and editing this manuscript. We acknowledge funding by the German Center of Infection Research (DZIF) TTU 08.708_00 to SH. Additionally, we acknowledge funding by the Deutsche Forschungsgemeinschaft DFG (German Research Foundation) in the frame of Germany´s Excellence Strategy – EXC 2124-390838134 (SH). Further, we acknowledge support by the Fortüne Program of the University Hospital Tübingen 2507-0-0 (SH). Work of DL was supported by Spanish Ministry of Science (PID2020-115699GB-100), and that of FG and SH was funded by the Deutsche Forschungsgemeinschaft (DFG, German Research Foundation) Project-ID 398967434-TRR261. We acknowledge support from the Open Access Publication Fund of the University of Tübingen.

## Additional information

### Funding

| Funder | Grant reference number | Author |
| --- | --- | --- |
| German Center of Infection Research | TTU 08.708_00 | Simon Heilbronner |
| Deutsche Forschungsgemeinschaft | EXC 2124 - 390838134 | Simon Heilbronner |
| Fortuene Program University Hospital Tuebingen | 2507-0-0 | Simon Heilbronner |
| Spanish Ministry of Science | PID2020-115699GB-100 | Daniel Lopez |
| Deutsche Forschungsgemeinschaft | 398967434 -TRR261 | Simon Heilbronner |

The funders had no role in study design, data collection and interpretation, or the decision to submit the work for publication.

### Author contributions

Lea Antje Adolf, Data curation, Investigation, Visualization, Methodology, Writing – original draft; Angelika Müller-Jochim, Lara Kricks, Supervision, Investigation, Methodology; Jan-Samuel Puls, Investigation, Visualization, Methodology; Daniel Lopez, Fabian Grein, Supervision, Methodology, Writing – review and editing; Simon Heilbronner, Conceptualization, Resources, Formal analysis, Supervision, Funding acquisition, Investigation, Visualization, Methodology, Project administration, Writing – review and editing

### Author ORCIDs

Jan-Samuel Puls  http://orcid.org/0000-0002-8130-7375
Daniel Lopez  http://orcid.org/0000-0002-8627-3813
Simon Heilbronner  http://orcid.org/0000-0002-6774-2311

### Decision letter and Author response

Decision letter https://doi.org/10.7554/eLife.85304.sa1
Author response https://doi.org/10.7554/eLife.85304.sa2

## Additional files

### Supplementary files

- MDAR checklist

### Data availability

All datasets underlying the diagrams are provided as Source Data.

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

# Appendix 1

## Appendix 1—key resources table

| Reagent type (species) or resource | Designation | Source or reference | Identifiers | Additional information |
|---|---|---|---|---|
| Strain, strain background (Escherichia coli) | BTH101 | BACTH System (Euromedex) | | Used for BACTH assay; F-, cya-99, araD139, galE15, galK16, rpsL1 (Str r), hsdR2, mcrA1, mcrB1 |
| Strain, strain background (Escherichia coli) | BTH101 pKT25+pUT18C | This study | | BACTH assay |
| Strain, strain background (Escherichia coli) | BTH101 pKT25-zip+pUT18C-zip | This study | | BACTH assay |
| Strain, strain background (Escherichia coli) | BTH101 pKT25:fhuB+pUT18C:fhuC | This study | | BACTH assay |
| Strain, strain background (Escherichia coli) | BTH101 pKT25:mntB+pUT18C:fhuC | This study | | BACTH assay |
| Strain, strain background (Escherichia coli) | BTH101 pKT25:isdF+pUT18C:fhuC | This study | | BACTH assay |
| Strain, strain background (Escherichia coli) | BTH101 pKT25:isdF_A213F+pUT18C:fhuC | This study | | BACTH assay |
| Strain, strain background (Escherichia coli) | BTH101 pKT25:isdF_G217F+pUT18C:fhuC | This study | | BACTH assay |
| Strain, strain background (Escherichia coli) | BTH101 pKT25:isdF_A+G_F+pUT18C:fhuC | This study | | BACTH assay |
| Strain, strain background (Escherichia coli) | BTH101 pKT25:mntB_CH_isdF +pU18C:fhuC | This study | | BACTH assay |
| Strain, strain background (Escherichia coli) | BTH101 pKT25:isdF_short_1+pUT18C:fhuC | This study | | BACTH assay |
| Strain, strain background (Escherichia coli) | BTH101 pKT25:isdF_short_2+pUT18C:fhuC | This study | | BACTH assay |
| Strain, strain background (Escherichia coli) | BTH101 pKT25:isdF_short_3+pUT18C:fhuC | This study | | BACTH assay |
| Strain, strain background (Escherichia coli) | BTH101 pKT25:isdF_short_4+pUT18C:fhuC | This study | | BACTH assay |
| Strain, strain background (Escherichia coli) | SA08B | Monk et al., 2015 | | Used for cloning of pIMAY, pRB474 and pRB473 plasmids; DC10B;0P$_{helpI}^-$ hsdMS (CC8-2) (SAUSA300_1751) of NRS384 integrated between the atpI and gidB genes |

*Appendix 1—key resources table continued on next page*

## Appendix 1—key resources table continued

| Reagent type (species) or resource | Designation | Source or reference | Identifiers | Additional information |
|---|---|---|---|---|
| Strain, strain background (Escherichia coli) | SLO1B | Heilbronner et al., 2013 | | Used for cloning of pIMAY for transformations of S. lugdunensis; DC10B hsdMS⁺ (S. lugdunensis N920143 (CC1)) |
| Strain, strain background (Staphylococcus aureus) | USA300 JE2 | Fey et al., 2013 | | WT |
| Strain, strain background (Staphylococcus aureus) | USA300 JE2 Δsbn | This study | | Markerless deletion of the entire sbn locus (staphyloferrin B biosynthesis genes; sbnA-I) |
| Strain, strain background (Staphylococcus aureus) | USA300 JE2 Δsfa | This study | | Markerless deletion of the entire sfa locus (staphyloferrin A biosynthesis genes; sfaA-D) |
| Strain, strain background (Staphylococcus aureus) | USA300 JE2 fhuC::Erm | Fey et al., 2013 | | Nebraska transposon library mutant ΔfhuC (SAUSA300_0633) |
| Strain, strain background (Staphylococcus aureus) | USA300 JE2 floA::Erm | Fey et al., 2013 | | Nebraska transposon library mutant ΔfloA (SAUSA300_1533) |
| Strain, strain background (Staphylococcus aureus) | USA300 JE2 floA_R | This study | | Genomic complementation of the Nebraska transposon library mutant floA::Erm with floA |
| Strain, strain background (Staphylococcus aureus) | COL | Gill et al., 2005 | | WT |
| Strain, strain background (Staphylococcus aureus) | Newman | Lorenz and Duthie, 1952 | | WT |
| Strain, strain background (Staphylococcus aureus) | Newman Δisd | This study | | Markerless deletion mutant of the entire isd locus |
| Strain, strain background (Staphylococcus aureus) | Newman Δfhu | This study | | Markerless deletion mutant of the entire fhu (fhuCBG) locus |
| Strain, strain background (Staphylococcus aureus) | Newman Δhts | This study | | Markerless deletion mutant of the entire hts (htsABC) locus |
| Strain, strain background (Staphylococcus aureus) | Newman Δsir | This study | | Markerless deletion mutant of the entire sir (sirABC) locus |
| Strain, strain background (Staphylococcus aureus) | Newman ΔfhuC | This study | | Markerless deletion mutant of fhuC |
| Strain, strain background (Staphylococcus aureus) | Newman ΔfloA | This study | | Markerless deletion mutant of floA |

*Appendix 1—key resources table continued*

| Reagent type (species) or resource | Designation | Source or reference | Identifiers | Additional information |
|---|---|---|---|---|
| Strain, strain background (Staphylococcus aureus) | Newman ΔcrtM | *Wieland et al., 1994* | | Deletion mutant of crtM by insertion of Cm resistance gene |
| Strain, strain background (Staphylococcus aureus) | Newman WT pRB473 | This study | | Empty vector control |
| Strain, strain background (Staphylococcus aureus) | Newman ΔfhuC pRB473 | This study | | Empty vector control |
| Strain, strain background (Staphylococcus aureus) | Newman ΔfhuC pRB473:fhuC | This study | | Expression plasmid for fhuC under its native promotor |
| Strain, strain background (Staphylococcus aureus) | Newman Δspa | This study | | Markerless deletion mutant of spa |
| Strain, strain background (Staphylococcus aureus) | Newman Δspa srtA::Erm | This study | | Phage transduction from the Nebraska transposon library mutant srtA::Erm (SAUSA300_2467) into Newman Δspa |
| Strain, strain background (Staphylococcus aureus) | Newman Δspa floA::Erm | This study | | Phage transduction from the Nebraska transposon library mutant floA::Erm (SAUSA300_1533) into Newman Δspa |
| Strain, strain background (Staphylococcus aureus) | Newman Δspa crtM::Erm | This study | | Phage transduction from the Nebraska transposon library mutant crtM::Erm (SAUSA300_2499) into Newman Δspa |
| Strain, strain background (Staphylococcus aureus) | Newman WT pRB474:isdF-3xFLAG | This study | | Expression plasmid for isdF-3xFLAG |
| Strain, strain background (Staphylococcus aureus) | Newman floA-6xHis pRB474:isdF-3xFLAG | This study | | Insertion of linker+6xHis-tag C-terminally of floA; expression plasmid for isdF-3xFLAG |
| Strain, strain background (Staphylococcus aureus) | Newman floA-SNAP pCQ11:isdF-mNeongreen | This study | | Insertion of SNAP tag C-terminally of floA; expression plasmid for isdF-mNeongreen |
| Strain, strain background (Staphylococcus aureus) | Newman ΔfloA pCQ11:isdF-mNeongreen | This study | | Markerless deletion of floA; expression plasmid for isdF-mNeongreen |
| Strain, strain background (Staphylococcus aureus) | Newman floA_R pCQ11:isdF-mNeongreen | This study | | Genomic complementation of the ΔfloA mutant; expression plasmid for isdF-mNeongreen |
| Strain, strain background (Staphylococcus lugdunensis) | N920143 | *Heilbronner et al., 2011* | | WT |
| Strain, strain background (Staphylococcus lugdunensis) | N920143 Δisd | *Zapotoczna et al., 2012* | | Markerless deletion mutant of the entire isd locus |

*Appendix 1—key resources table continued on next page*

*Appendix 1—key resources table continued*

| Reagent type (species) or resource | Designation | Source or reference | Identifiers | Additional information |
|---|---|---|---|---|
| Strain, strain background (*Staphylococcus lugdunensis*) | N920143 Δ*lha* | **Jochim et al., 2020** | | Markerless deletion mutant of *lhaSTA* |
| Strain, strain background (*Staphylococcus lugdunensis*) | N920143 Δ*lha*Δ*floA* | This study | | Markerless deletion mutant of *lhaSTA* and *floA* |
| Biological sample (*Human*) | Human hemoglobin | Own purification (see Materials and methods) | | Sex male |
| Antibody | Mouse monoclonal anti-FLAG M2 | Sigma | F3165 | WB (1:10,000) |
| Antibody | Polyclonal IRDye 800CW goat anti-mouse IgG secondary antibody | LI-COR | 926-32210 | WB (1:10,000) |
| Antibody | Polyclonal rabbit serum anti-IsdA | Prof. J. Geoghegan | | WB (1:5000) |
| Antibody | Polyclonal IRDye 680RD goat anti-rabbit IgG secondary antibody | LI-COR | 926-68071 | WB (1:10,000) |
| Recombinant DNA reagent | pIMAY (plasmid) | **Monk et al., 2012** | | *E. coli*/*Staphylococcus* thermo-sensitive vector for allelic replacement in *S. aureus* |
| Recombinant DNA reagent | pIMAY:Δ*sbn* | This study | | Plasmid for the deletion of the entire *sbn* locus |
| Recombinant DNA reagent | pIMAY:Δ*sfa* | This study | | Plasmid for the deletion of the entire *sfa* locus |
| Recombinant DNA reagent | pIMAY:Δ*floA* complementation | This study | | Plasmid for the genomic reversion of in USA300 JE2 *floA*::*Erm* and Newman Δ*floA* |
| Recombinant DNA reagent | pIMAY:Δ*isd* | This study | | Plasmid for the deletion of the entire *isd* locus |
| Recombinant DNA reagent | pIMAY:Δ*fhu* | This study | | Plasmid for the deletion of the entire *fhu* locus |
| Recombinant DNA reagent | pIMAY:Δ*hts* | This study | | Plasmid for the deletion of the entire *hts* locus |
| Recombinant DNA reagent | pIMAY:Δ*sir* | This study | | Plasmid for the deletion of the entire *sir* locus |
| Recombinant DNA reagent | pIMAY:Δ*fhuC* | This study | | Plasmid for the deletion of *fhuC* |

*Appendix 1—key resources table continued on next page*

## Appendix 1—key resources table continued

| Reagent type (species) or resource | Designation | Source or reference | Identifiers | Additional information |
|---|---|---|---|---|
| Recombinant DNA reagent | pIMAY:ΔfloA | This study | | Plasmid for the deletion of floA |
| Recombinant DNA reagent | pIMAY:Δspa | This study | | Plasmid for the deletion of spa |
| Recombinant DNA reagent | pIMAY:ΔfloA | This study | | Plasmid for the deletion of floA in S. lugdunensis N920143 |
| Recombinant DNA reagent | pIMAY:floA-6xHis | This study | | Plasmid for the addition of 6xHis C-terminally to floA |
| Recombinant DNA reagent | pIMAY:floA-SNAP | This study | | Plasmid for the addition of SNAP C-terminally to floA |
| Recombinant DNA reagent | pRB473 (plasmid) | Brückner, 1992 | | Expression plasmid without a promotor |
| Recombinant DNA reagent | pRB473:fhuC | This study | | fhuC expressing plasmid under its native promotor (fur box) |
| Recombinant DNA reagent | pRB474 (plasmid) | Brückner, 1992 | | Expression plasmid with constitutive promotor |
| Recombinant DNA reagent | pRB474:isdF-3xFLAG | This study | | isdF-3xFLAG expressing plasmid |
| Recombinant DNA reagent | pCQ11:snap | Lund et al., 2018 | | Used as template for SNAP amplification PCR |
| Recombinant DNA reagent | pCQ11:gfp | C. Quiblier and B. Berger-Bächi | | Backbone for pCQ11:mNeongreen |
| Recombinant DNA reagent | pCQ11:mNeongreen | This study | | Backbone for pCQ11 :isdF-mNeongreen |
| Recombinant DNA reagent | pLOM-S-mNeongreen-EC18153 | Addgene plasmid | # 137075 | mNeongreen template |
| Recombinant DNA reagent | pCQ11:isdF-mNeongreen | This study | | isdF- with C-terminal mNeongreen fusion; IPTG inducible |
| Recombinant DNA reagent | pKT25 (plasmid) | BACTH System (Euromedex) | | BACTH assay plasmid, N-terminal T25 fragment |
| Recombinant DNA reagent | pKT25:fhuB | This study | | T25 fragment N-terminally of fhuB |
| Recombinant DNA reagent | pKT25:isdF | This study | | T25 fragment N-terminally of isdF |
| Recombinant DNA reagent | pKT25:mntB | This study | | T25 fragment N-terminally of mntB |

*Appendix 1—key resources table continued on next page*

*Appendix 1—key resources table continued*

| Reagent type (species) or resource | Designation | Source or reference | Identifiers | Additional information |
|---|---|---|---|---|
| Recombinant DNA reagent | pKT25:zip | BACTH System (Euromedex) | | T25 fragment N-terminally of *zip*; positive control |
| Recombinant DNA reagent | pKT25:isdF_short1 | This study | | T25 fragment N-terminally of *isdF_short1*: truncated C-terminus |
| Recombinant DNA reagent | pKT25:isdF_short2 | This study | | T25 fragment N-terminally of *isdF_short2*: truncated C-terminus+fourth cytosolic loop |
| Recombinant DNA reagent | pKT25:isdF_short3 | This study | | T25 fragment N-terminally of *isdF_short3*: truncated C-terminus+third cytosolic loop |
| Recombinant DNA reagent | pKT25:isdF_short4 | This study | | T25 fragment N-terminally of *isdF_short4*: truncated C-terminus+second cytosolic loop |
| Recombinant DNA reagent | pKT25:isdF_A213F | This study | | T25 fragment N-terminally of *isdF*: alanine position 213 exchanged to phenylalanine |
| Recombinant DNA reagent | pKT25:isdF_G217F | This study | | T25 fragment N-terminally of *isdF*: glycine position 217 exchanged to phenylalanine |
| Recombinant DNA reagent | pKT25:isdF_A+G_F | This study | | T25 fragment N-terminally of *isdF*: alanine (213)+glycine (217) exchanged to phenylalanines |
| Recombinant DNA reagent | pKT25:mntB_CHisdF | This study | | T25 fragment N-terminally of *mntB*; coupling helix of *mntB* exchanged to the one from *isdF* |
| Recombinant DNA reagent | pUT18C | BACTH System (Euromedex) | | BACTH assay plasmid, N-terminal T18 fragment |
| Recombinant DNA reagent | pUT18C:fhuC | This study | | T18 fragment N-terminally of *fhuC* |
| Recombinant DNA reagent | pUT18C:zip | BACTH System (Euromedex) | | T18 fragment N-terminally of *zip*; positive control |
| Recombinant DNA reagent | pRB474:mprFdelCysflag | *Slavetinsky et al., 2022* | | Used as template for 3xFLAG amplification PCR |
| Sequence-based reagent | Δsbn_PF-A_SmaI | This study | PCR primer | Primer for Δ*sbnA-I* fragment using pIMAY; CACCTAAAGATCCCGGGACGTCAGTGGC |
| Sequence-based reagent | Δsbn_PR-B | This study | PCR primer | Primer for Δ*sbnA-I* fragment using pIMAY; CATAGGTGTTTGCCCTACAGAATCTAAC |
| Sequence-based reagent | Δsbn_PF-C | This study | PCR primer | Primer for Δ*sbnA-I* fragment using pIMAY; CTGTAGGGCAAACACCTATGTAGTTTT ACTGTGATGTTGAGGGAAATA |

Appendix 1—key resources table continued

| Reagent type (species) or resource | Designation | Source or reference | Identifiers | Additional information |
|---|---|---|---|---|
| Sequence-based reagent | Δsbn_PR-D_KpnI | This study | PCR primer | Primer for ΔsbnA-I fragment using pIMAY; AAATCAGCAAGGTACCACCAATCAGCC |
| Sequence-based reagent | ΔsfaDABC_PF-A_KpnI | This study | PCR primer | Primer for ΔsfaDABC fragment using pIMAY; GATCGGTACCAGTATCTTTAGTTGATGATTCT |
| Sequence-based reagent | ΔsfaDABC_PR-B | This study | PCR primer | Primer for ΔsfaDABC fragment using pIMAY; TAATATATTTATCAATAAGTCTAAGTTGACA |
| Sequence-based reagent | ΔsfaDABC_PF-C | This study | PCR primer | Primer for ΔsfaDABC fragment using pIMAY; ACTTATTGATAAATATATTATAAGGT TATAGAATTTTATTAATCGT |
| Sequence-based reagent | ΔsfaDABC_PR-D | This study | PCR primer | Primer for ΔsfaDABC fragment using pIMAY; CGGAATTCTTCTATTGGTAGTGTAAGTTGGATCA |
| Sequence-based reagent | ΔfloA_PF-A_SacI | This study | PCR primer | Primer for floA::Erm and ΔfloA complementation fragment using pIMAY (floA_R); CCAAGGAGCTCTCAAATATGCATTCTATC |
| Sequence-based reagent | ΔfloA comp._PR-B_SmaI | This study | PCR primer | Primer for floA::Erm and ΔfloA complementation fragment using pIMAY (floA_R); CTTCACCCAACCCGGGCGATGATTGTTTC |
| Sequence-based reagent | ΔfloA comp_PF-C_SmaI | This study | PCR primer | Primer for floA::Erm and ΔfloA complementation fragment using pIMAY (floA_R); GAAACAATCATCGCCCGGGTTGGTGAAG |
| Sequence-based reagent | ΔfloA_PR-D_KpnI | This study | PCR primer | Primer for floA::Erm and ΔfloA complementation fragment using pIMAY (floA_R); TTTTCGGTACCAATGTCAGTACGAATC |
| Sequence-based reagent | Δisd_PF-A_KpnI | This study | PCR primer | Primer for ΔisdB-G fragment using pIMAY; TAAAGGGAACAAAAGCTGGGTACCAT GCAGAGGACTTACTTGCGTAAAG |
| Sequence-based reagent | Δisd_PR-B | This study | PCR primer | Primer for ΔisdB-G fragment using pIMAY; TAAATTAACAAATTTTAATTGGCGGATG |
| Sequence-based reagent | Δisd_PF-C | This study | PCR primer | Primer for ΔisdB-G fragment using pIMAY; ATTAAAATTTGTTAATTTAAGAATTTAAA GAGGTTGCAGTACTTGTTATG |
| Sequence-based reagent | Δisd_PR-D_SacI | This study | PCR primer | Primer for ΔisdB-G fragment using pIMAY; CGACTCACTATAGGGCGAATTGGAGCTC TCAATTAAATGCACACCTTCAATTAAAGC |

Appendix 1—key resources table continued

| Reagent type (species) or resource | Designation | Source or reference | Identifiers | Additional information |
|---|---|---|---|---|
| Sequence-based reagent | Δfhu_PF-A_SacI | This study | PCR primer | Primer for ΔfhuCBG fragment using pIMAY; AATACCTCGAGCTCAGCACGCCATATG CTTTGCTTTCTTCGAT |
| Sequence-based reagent | Δfhu_PR-B | This study | PCR primer | Primer for ΔfhuCBG fragment using pIMAY; CATAATTTCCCTACTTTCAATAAAATTCTT |
| Sequence-based reagent | Δfhu_PF-C | This study | PCR primer | Primer for ΔfhuCBG fragment using pIMAY; ATTTTATTGAAAGTAGGGAAATTATG TAGTGTCAATGGACACAACTTATTGCTATG |
| Sequence-based reagent | Δfhu_PR-D_KpnI | This study | PCR primer | Primer for ΔfhuCBG fragment using pIMAY; TGCTTTggTAcCTTCTAAATATTTATCAGGTGTAGG |
| Sequence-based reagent | Δhts_PF-A_SacI | This study | PCR primer | Primer for ΔhtsABC fragment using pIMAY; GCACgagCTCATTCGATGTATGAAAAATTTAC |
| Sequence-based reagent | Δhts_PR-B | This study | PCR primer | Primer for ΔhtsABC fragment using pIMAY; CATCGTTCCACTCCTTAATATGTATAAC |
| Sequence-based reagent | Δhts_PF-C | This study | PCR primer | Primer for ΔhtsABC fragment using pIMAY; TATACATATTAAGGAGTGGAACGATGTA ACTAACATATGATTAGAGTTTAAAAAAG |
| Sequence-based reagent | Δhts_PR-D_KpnI | This study | PCR primer | Primer for ΔhtsABC fragment using pIMAY; GTCAGGTacCAATTTATCTTTTAAAATAG |
| Sequence-based reagent | Δsir_PF-A_SacI | This study | PCR primer | Primer for ΔsirABC fragment using pIMAY; GTTTTgAgCtCTTGATTTTAGCTATCATTG |
| Sequence-based reagent | Δsir_PR-B | This study | PCR primer | Primer for ΔsirABC fragment using pIMAY; CATTGACTAATTAGCCTCCTTCGTG |
| Sequence-based reagent | Δsir_PF-C | This study | PCR primer | Primer for ΔsirABC fragment using pIMAY; AGGAGGCTAATTAGTCAATGTAACG ATATTATTAAAACAAAATG |
| Sequence-based reagent | Δsir_PR-D_KpnI | This study | PCR primer | Primer for ΔsirABC fragment using pIMAY; CTGATGgtAccAATAAGTCAGTAATATAAAATTC |
| Sequence-based reagent | PF-A_ΔfhuC_SacI | This study | PCR primer | Primer for ΔfhuC fragment using pIMAY; AATACCTCGAGCTCAGCACGCCATAT GCTTTGCTTTCTTCGAT |
| Sequence-based reagent | PR-B_ΔfhuC | This study | PCR primer | Primer for ΔfhuC fragment using pIMAY; CATAATTTCCCTACTTTCAATAAAATTCTT |

Appendix 1—key resources table continued

| Reagent type (species) or resource | Designation | Source or reference | Identifiers | Additional information |
|---|---|---|---|---|
| Sequence-based reagent | PF-C_ΔfhuC | This study | PCR primer | Primer for ΔfhuC fragment using pIMAY; TTATTGAAAGTAGGGAAATTAT GTAATTAAGTAAGTAAATAT |
| Sequence-based reagent | PR-D_ΔfhuC_KpnI | This study | PCR primer | Primer for ΔfhuC fragment using pIMAY; ATGGTAAGTTGGGTACCCAATTGTTAA TATAATGAATAACGCAATACCA |
| Sequence-based reagent | ΔfloA_PF-A_SacI | This study | PCR primer | Primer for ΔfloA fragment using pIMAY; CCAAGGAGCTCTCAATATGCATTCTATC |
| Sequence-based reagent | ΔfloA_PR-B | This study | PCR primer | Primer for ΔfloA fragment using pIMAY; AAACATGGTATCGCTCCTTTTAATTAATC |
| Sequence-based reagent | ΔfloA_PF-C | This study | PCR primer | Primer for ΔfloA fragment using pIMAY; AAAGGAGCGATACCCATGTTTAAGT CGAGAGGTGATTAAATG |
| Sequence-based reagent | ΔfloA_PR-D_KpnI | This study | PCR primer | Primer for ΔfloA fragment using pIMAY; TTTTCGGTACCAATGTCAGTACCGAATC |
| Sequence-based reagent | Δspa_PF-A_SacI | This study | PCR primer | Primer for Δspa fragment using pIMAY; GAAAGAGCTCTTTAATTCATATGGATGAC |
| Sequence-based reagent | Δspa_PR-B | This study | PCR primer | Primer for Δspa fragment using pIMAY; CATAATATAACGAATTATGTATTGCAATAC |
| Sequence-based reagent | Δspa_PF-C | This study | PCR primer | Primer for Δspa fragment using pIMAY; ACATAATTCGTTATATTATGTAAAAAC AAACAATACACAACGATAG |
| Sequence-based reagent | Δspa_PR-D_KpnI | This study | PCR primer | Primer for Δspa fragment using pIMAY; CAGGTGGGGTACCAGCGAAACTTATTTCAC |
| Sequence-based reagent | N9_PF-A_ΔfloA_SacI | This study | PCR primer | Primer for ΔfloA (for S. lugdunensis N920143) fragment using pIMAY; CTTTATTGGAGCTCCAGTAATAGGCTTTTTGGCATAG |
| Sequence-based reagent | N9_PR-B_ΔfloA | This study | PCR primer | Primer for ΔfloA (for S. lugdunensis N920143) fragment using pIMAY; CATTAAATCACTCCTATAAATTAAATCTATC |
| Sequence-based reagent | N9_PF-C_ΔfloA | This study | PCR primer | Primer for ΔfloA (for S. lugdunensis N920143) fragment using pIMAY; TTTATAGGAGTGATTAAATGTAATTA AAGGGGTGATGTCATGAAC |

## Appendix 1—key resources table continued

| Reagent type (species) or resource | Designation | Source or reference | Identifiers | Additional information |
|---|---|---|---|---|
| Sequence-based reagent | N9_PR_ΔfloA_KpnI | This study | PCR primer | Primer for ΔfloA (for S. lugdunensis N920143) fragment using pIMAY; CGAACAGGTACCAAATCATCCATAAGTGTATGTTC |
| Sequence-based reagent | PF-A_FloA-6xHis_SacI | This study | PCR primer | Primer for floA-6xHis fragment including linker+6xHis using pIMAY; ATATTGagCtcCTTGTTGGTGGTGCTGGTGAAGAAAC |
| Sequence-based reagent | PR-B_FloA-6xHis | This study | PCR primer | Primer for floA-6xHis fragment including linker+6xHis using pIMAY; GTGATGGTGATGGTGATGCGATCCTCTATGTTCAGGTGACTCATCATCACTTTG |
| Sequence-based reagent | PF-C_FloA-6xHis | This study | PCR primer | Primer for floA-6xHis fragment including linker+6xHis using pIMAY; CATAGAGGATCGCATCACCATCACCATCACTAAGTCGAGAGGTGATTAAATGAGTG |
| Sequence-based reagent | PR-D_FloA-6xHis_KpnI | This study | PCR primer | Primer for floA-6xHis fragment including linker+6xHis using pIMAY; TTTTCggTAcCAATGTCAGTACGAATCGTTTTAATATC |
| Sequence-based reagent | PF-A_FloA-SNAP_SacI | This study | PCR primer | Primer for floA-SNAP fragment using pIMAY; ATATTGagCtcCTTGTTGGTGGTGCTGGTGAAGAAAC |
| Sequence-based reagent | PR-B_FloA-SNAP | This study | PCR primer | Primer for floA-SNAP fragment using pIMAY; ATTTCGCAATCTTTGTCCATATGTTCAGGTGACTCATCATCACTTTG |
| Sequence-based reagent | PF-SNAP | This study | PCR primer | Primer for floA-SNAP fragment using pIMAY; ATGGACAAAGATTGCGAAATGAAACG |
| Sequence-based reagent | PR-SNAP | This study | PCR primer | Primer for floA-SNAP fragment using pIMAY; TCATCCCAGACCCGGTTTACCCAG |
| Sequence-based reagent | PF-C_FloA-SNAP | This study | PCR primer | Primer for floA-SNAP fragment using pIMAY; GTAAACCGGGTCTGGGATGAGTCGAGAGGTGATTAAATGAGTG |
| Sequence-based reagent | PR-D_FloA-SNAP_KpnI | This study | PCR primer | Primer for floA-SNAP fragment using pIMAY; TTTTCggTAcCAATGTCAGTACGAATCGTTTTAATATC |
| Sequence-based reagent | mNeon-for (NheI, SmaI) | This study | PCR primer | Primer for isdF-mNeongreen construct; TTATGCTAGCTTAACCCGGGATGGCGTCGAAGG |
| Sequence-based reagent | mNeon-rev (AscI) | This study | PCR primer | Primer for isdF-mNeongreen construct; TATAGGCGCGCCTCAACCTCCTTTATAGAG |
| Sequence-based reagent | isdF-for (NheI) | This study | PCR primer | Primer for isdF-mNeongreen construct; GCTCGGCTAGCATGATGATAAAAAATAAAAAG |

*Appendix 1—key resources table continued on next page*

# Appendix 1—key resources table continued

| Reagent type (species) or resource | Designation | Source or reference | Identifiers | Additional information |
|---|---|---|---|---|
| Sequence-based reagent | isdF-rev (SmaI) | This study | PCR primer | Primer for isdF-mNeongreen construct; AATATCCCGGGGATTCGATTTCGTTGAC |
| Sequence-based reagent | pcq11-seq2-for | This study | PCR primer | Primer for isdF-mNeongreen construct; GTTGACTTTATCTACAAGG |
| Sequence-based reagent | pcq11-seq2-rev | This study | PCR primer | Primer for isdF-mNeongreen construct; TCTCGAAAATAATAGAGGG |
| Sequence-based reagent | PF_furbox + fhuC_SacI | This study | PCR primer | Primer for cloning of fur box+fhuC into pRB473; AAAAGAGCTCTTAGTCAATAAGATTG |
| Sequence-based reagent | PR_furbox + fhuC_HindIII | This study | PCR primer | Primer for cloning of fur box+fhuC into pRB473; ATTAACAAGCTTAATTAAGAATAAGCTCTG |
| Sequence-based reagent | PF_474-RBS-IsdF_PstI | This study | PCR primer | Primer for cloning isdF-3xFLAG into pRB474; ATGCCTGCAGaggaggattagttATGA TGATAAAAATAAAAAGAAACTAC |
| Sequence-based reagent | PR_474-IsdF | This study | PCR primer | Primer for cloning isdF-3xFLAG into pRB474; CCGTCATGGTCTTTGTAGTCGAT TCGATTTCGTTGACTTTGAC |
| Sequence-based reagent | PF-3xFLAG | This study | PCR primer | Primer for cloning isdF-3xFLAG into pRB474; GACTACAAAGACCATGACGGTGATTAT |
| Sequence-based reagent | PR-474-3xFLAG_SacI | This study | PCR primer | Primer for cloning isdF-3xFLAG into pRB474; TCTATgagctcTCATTTGTCATCGTCATCCTTg |
| Sequence-based reagent | PF_FhuB_pKT25_PstI | This study | PCR primer | Primer for cloning fhuB into pKT25; AAAACTGCAGTTAACATGACAAATA |
| Sequence-based reagent | PR_FhuB_pKT25_EcoRI | This study | PCR primer | Primer for cloning fhuB into pKT25; TGCGTGAATTCTTTGAACTAATCATAT |
| Sequence-based reagent | PF_isdF_pKT25_PstI | This study | PCR primer | Primer for cloning isdF into pKT25; GGATAAAAATCTGCAGTTGATATGATGATA |
| Sequence-based reagent | PR_isdF-pKT25_EcoRI | This study | PCR primer | Primer for cloning isdF into pKT25; CACTAAACCAGGAATTCTACCGTTTTAGAT |
| Sequence-based reagent | MntB_KT-PF_PstI | This study | PCR primer | Primer for cloning mntB into pKT25; TAGTCAAAGGCTGCAGATAACATGTTAG |
| Sequence-based reagent | MntB_PR_EcoRI | This study | PCR primer | Primer for cloning mntB into pKT25; CTAATAATAAAGGTACTgAaTTcTcATG |
| Sequence-based reagent | PF_pKT_SmaI | This study | PCR primer | Primer for cloning mntB into pKT25; GAAAACCCGGGCGTTACCCAACTTAATC |

*Appendix 1—key resources table continued on next page*

**Appendix 1—key resources table continued**

| Reagent type (species) or resource | Designation | Source or reference | Identifiers | Additional information |
|---|---|---|---|---|
| Sequence-based reagent | PR_pKT_stop_SmaI | This study | PCR primer | Primer for cloning *mntB* into pKT25; CCAGCCCGGGCGTTGTAAAACTACGG |
| Sequence-based reagent | PF_IsdF_short_after MCS_KpnI | This study | PCR primer | Primer for cloning *isdF_short1/2/3/4* into pKT25; GTAGGGTACCGCCGTAGTTTTACAAC |
| Sequence-based reagent | PR_IsdF_short_1_KpnI | This study | PCR primer | Primer for cloning *isdF_short1* into pKT25; TTGAggTaccCAAATTAAGTAAATTAG |
| Sequence-based reagent | PR_IsdF_short_2_KpnI | This study | PCR primer | Primer for cloning *isdF_short2* into pKT25; GTAAggtaCCCCAACTAGCTTTCTAAC |
| Sequence-based reagent | PR_IsdF_short_3_KpnI | This study | PCR primer | Primer for cloning *isdF_short3* into pKT25; TAGTAAAggtAccTTAGGGGACAATAG |
| Sequence-based reagent | PR_IsdF_short_4_KpnI | This study | PCR primer | Primer for cloning *isdF_short4* into pKT25; AGAAgGtacCAATATAATTATTAAAAATGG |
| Sequence-based reagent | PF_KT-IsdF_A213F | This study | PCR primer | Primer for cloning *isdF_A213F* into pKT25; CGACATACAAttCGAAGTATCGGTTTTA ATATTGATCGTTACAGATGG |
| Sequence-based reagent | PR_KT-IsdF_A213F | This study | PCR primer | Primer for cloning *isdF_A213F* into pKT25; CCATCTGTAACGATCAATATTAAAAACC GATACTTCGaaaTTGTATGTCG |
| Sequence-based reagent | PF_KT-IsdF_G217F | This study | PCR primer | Primer for cloning *isdF_G217F* into pKT25; CGACATACAAGCGCGAAGTATCttTTTTA ATATTGATCGTTACAGATGG |
| Sequence-based reagent | PR_KT-IsdF_G217F | This study | PCR primer | Primer for cloning *isdF_G217F* into pKT25; CCATCTGTAACGATCAATATTAAAAaaG ATACTTCGCGCTTGTATGTCG |
| Sequence-based reagent | PF_KT-IsdF_A+G_F | This study | PCR primer | Primer for cloning *isdF_A+G_F* into pKT25; CGACATACAAttCGAAGTATCttTTTTA ATATTGATCGTTACAGATGG |
| Sequence-based reagent | PR_KT-IsdF_A+G_F | This study | PCR primer | Primer for cloning *isdF_A+G_F* into pKT25; CCATCTGTAACGATCAATATTAAAAaa GATACTTCGaaaTTGTATGTCG |
| Sequence-based reagent | MntB_KT-P-F (PstI) | This study | PCR primer | Primer for cloning *mntB_CH*~IsdF~ into pKT25; TAGTCAAAGGCTGCAGATAAACATGTTAG |
| Sequence-based reagent | PR_MntB-CHisdF | This study | PCR primer | Primer for cloning *mntB_CH*~IsdF~ into pKT25; ACCGATACTTCGCGCTTGTATGTCGTCT AAATTTAGTAAAATTATAGAAAATAATGAT TAGAATAAGGACGATTGAACCAATCAC |

*Appendix 1—key resources table continued on next page*

*Appendix 1—key resources table continued*

| Reagent type (species) or resource | Designation | Source or reference | Identifiers | Additional information |
|---|---|---|---|---|
| Sequence-based reagent | PF_MntB-CHisdF | This study | PCR primer | Primer for cloning mntB_CH_isdF into pKT25; AATTTACTAAATTTAGACGACACAAGCG CGAAGTATCGGTACGACGTTATTACATTAC TTTGTGATGTTGTTACTCTCATTAG |
| Sequence-based reagent | PR_pKT_stop_SmaI | This study | PCR primer | Primer for cloning mntB_CH_isdF into pKT25; CCAGCCCGGCGCGTTGTAAAACTACGG |
| Sequence-based reagent | PF_FhuC-SalI | This study | PCR primer | Primer for cloning fhuC into pUT18C; AGAAAGAAGTCGACTGAAAGTAGGGAAATTATG |
| Sequence-based reagent | PR_FhuC_EcoRI | This study | PCR primer | Primer for cloning fhuC into pUT18C; TATTGAATTCCTTAATTAAGAATAAGCTCT |
| Commercial assay or kit | BACTH System Kit | Euromedex | Cat. No.: EUK001 | |
| Commercial assay or kit | CelLytic MEM protein extraction Kit | Sigma | CE0050 | |
| Chemical compound, drug | RPMI-1640 medium | Sigma | R6504-10L | |
| Chemical compound, drug | Bacto casamino acids | BD Biosciences | 223050 | |
| Chemical compound, drug | EDDHA | LGC Standards | TRC-E335100-10MG | |
| Chemical compound, drug | Zaragozic acid A trisodium salt | Santa Cruz Biotechnology | SC-302001 | |
| Chemical compound, drug | ONPG | Sigma | N1127 | |
| Chemical compound, drug | DDM | Roth | CN26.1 | |
| Chemical compound, drug | Profinity IMAC Resin Ni-charged | Bio-Rad | 1560135 | |
| Other | Ferrichrome | EMC Microcollections | FCH | See Materials and methods: Growth in iron-limited medium |
| Other | Aerobactin | EMC Microcollections | Fe-AERO | See Materials and methods: Growth in iron-limited medium |
| Other | SNAP-Cell TMR-Star | New England Biolabs | S9105S | Materials and methods: Fluorescence microscopy |

