## [Editor Report]

In this fundamental manuscript, the authors provide compelling evidence that a housekeeping ATPase is required for heme utilization in the important pathogen *Staphylococcus aureus* through its interaction with the canonical heme transporter in this organism. The authors convincingly show that this complex associates with functional membrane microdomains and thus establishes a new paradigm for regional localization of the heme transport system in the staphylococci. The work will be of interest to microbiologists, particularly those studying transport for macromolecules.

---

## [Decision Letter]

**Decision letter after peer review:**

Thank you for submitting your article "Functional membrane microdomains and the hydroxamate siderophore transporter ATPase FhuC govern Isd-dependent heme acquisition in *Staphylococcus aureus*" for consideration by *eLife*. Your article has been reviewed by 2 peer reviewers, and the evaluation has been overseen by a Reviewing Editor and Bavesh Kana as the Senior Editor. The following individual involved in the review of your submission has agreed to reveal their identity: Eric Skaar (Reviewer #1).

Essential revisions:

1. The manuscript convincingly demonstrates that FhuC is required for heme iron utilization and presents strong data to implicate FhuC in binding to IsdF. The authors report that IsdF localizes to functional membrane microdomains in *S. aureus*. These experiments would benefit from controls showing that the DRM fraction contains the functional membrane microdomains and that the fractionation was successful.

2. The authors also present strong data demonstrating that loss of floA prevents IsdF incorporation into the membrane although these data would also benefit from genetic complementation.

3. In a surprising result, the authors report that the IsdA protein is not localized in the functional membrane microdomains which are confounding since IsdA is modeled to work in concert with IsdF. These data suggest there is much more to learn regarding the spatial distribution of this transport system. Further discussion on this would be useful.

4. More details concerning different strategies of iron acquisition should be mentioned in the introduction.

5. Additional bibliographic literature is needed for explaining what unknown ATPase partially substitutes for the function of FhuC.

6. Figure 8 shows the proposed model of heme funneling over the *S. aureus* cell envelope should be enhanced and legends must be added.

7. Compare, if possible, the growth of both mutant and wild-type strains in the presence of both FeSO4 and hHb.

*Reviewer #2 (Recommendations for the authors):*

– Figure 8 shows the proposed model of heme funneling over the *S. aureus* cell envelope should be enhanced and legends must be added

– Specify the chemicals used.

– It is interesting to compare, if possible, the growth of both mutant and wild-type strains in the presence of both FeSO4 and hHb.

---

## [Author Response]

Essential revisions:1. The manuscript convincingly demonstrates that FhuC is required for heme iron utilization and presents strong data to implicate FhuC in binding to IsdF. The authors report that IsdF localizes to functional membrane microdomains in *S. aureus*. These experiments would benefit from controls showing that the DRM fraction contains the functional membrane microdomains and that the fractionation was successful.

The disruption of biological membranes using nonionic detergents followed by division into DRM and DSM fractions is a method developed to investigate eukaryotic lipid rafts and used for decades. Similarly, functional membrane microdomains (FMMs) and their protein components concentrate in the DRM fraction due to their compact and increased hydrophobic compositions, which makes them more resistant to the detergent treatment. This method is used consistently by scientists investigating FMMs in multiple species (Bramkamp & Lopez, 2015; Brown, 2002; Shah & Sehgal, 2007).

It has to be emphasized that the DRM fraction is not identical to FMMs (Hancock, 2006; Lichtenberg et al., 2005). Proteins that are not associated with FMMs in living cells might accumulate in the DRM fraction upon membrane disruption and vice versa. Accordingly, enrichment of a protein of interest in the DRM fraction is by no means proof for its association with FMMs in vivo or for functional relevance of this association. Our observation that the activity of sortase A is independent of FMM formation despite their association with the DRM fraction highlights this problem. However, the method can serve as a starting point for further experiments.

We did not perform additional experiments to confirm the successful application of the DRM/DSM technology for three reasons.

1) The methodology is highly standardized (commercially available kit).

2) Our results that IsdF accumulates in the DRM fraction are congruent with results previous published by others (Garcia-Fernandez et al., 2017).

3) The method is not particularly strong evidence and conversely discussed additional controls would not provide clarification. Therefore, we validated our conclusions with a substantial set of downstream experiments using different approaches. We performed FloA-pull down assay to study protein-protein interaction of FloA and IsdF (Figure 3 C) and co-localization microscopy (Figure 4). Altogether confirming the targeted association of IsdF with FMMs in living cells.

2. The authors also present strong data demonstrating that loss of floA prevents IsdF incorporation into the membrane although these data would also benefit from genetic complementation.

We agree! We now created a genetic revertant by replacing the ΔfloA mutant allele with the functional WT-allele at the same chromosomal position (Newman floA_R). The repaired strain showed (WT-levels of IsdF signal) in microscopy experiments (Figure 4 C+D). The new data was added to the Results section (Ln 205-207).

3. In a surprising result, the authors report that the IsdA protein is not localized in the functional membrane microdomains which are confounding since IsdA is modeled to work in concert with IsdF. These data suggest there is much more to learn regarding the spatial distribution of this transport system. Further discussion on this would be useful.

I think this is a misunderstanding. IsdA is anchored to the cell wall by sortase A and will therefore not be directly associated with FMMs. However, appropriate sorting of IsdA to the cell wall is essential for heme funneling to the membrane and the responsible enzyme (Sortase A) is found enriched in FMMs. Therefore, we tested if FMM-deficiency corrupts the entire Isd-funnel including sorting of IsdA. However, IsdA could still be detected in the cell wall fraction in the FMM mutants, suggesting that sortase A remained functional. This leads us to conclude that the cell wall-associated heme-funnel is most likely intact in FMM-mutants and the observed growth defect in the presence of hemoglobin must be due to inappropriate membrane location of IsdF.

We fully agree, there is a lot to learn about how the cell wall-funnel and the membrane-associated transporter need to be spatially organized to optimize heme shuttling from the cell wall to the membrane. We can only speculate about this and use the “Speculation” paragraph (Ln 372-391) and Figure 9 for this purpose.

4. More details concerning different strategies of iron acquisition should be mentioned in the introduction.

We included information about siderophore iron and Feo transporters in the introduction (Ln 51-61).

5. Additional bibliographic literature is needed for explaining what unknown ATPase partially substitutes for the function of FhuC.

We added information regarding additional iron regulated ATPases that might partially substitute for FhuC to the discussion (Ln 304-308).

6. Figure 8 shows the proposed model of heme funneling over the *S. aureus* cell envelope should be enhanced and legends must be added.

We enhanced former Figure 8, now Figure 9. We increased the text size and included a legend.

7. Compare, if possible, the growth of both mutant and wild-type strains in the presence of both FeSO4 and hHb.

We now show the FeSO_4_ controls for WT and mutant strains, which can be found in Figure 5-supplement 1, Figure 6-supplement 1 and Figure 8 C. In none of the experiments we observed differences between WT and mutants.

References:

Bramkamp, M., & Lopez, D. (2015). Exploring the existence of lipid rafts in bacteria. Microbiol Mol Biol Rev, 79(1), 81-100. https://doi.org/10.1128/MMBR.00036-14

Brown, D. A. (2002). Isolation and use of rafts. Curr Protoc Immunol, Chapter 11, Unit 11.10. https://doi.org/10.1002/0471142735.im1110s51

Garcia-Fernandez, E., Koch, G., Wagner, R. M., Fekete, A., Stengel, S. T., Schneider, J., Mielich-Suss, B., Geibel, S., Markert, S. M., Stigloher, C., & Lopez, D. (2017). Membrane Microdomain Disassembly Inhibits MRSA Antibiotic Resistance. Cell, 171(6), 1354-1367 e1320. https://doi.org/10.1016/j.cell.2017.10.012

Hancock, J. F. (2006). Lipid rafts: contentious only from simplistic standpoints. Nature Reviews Molecular Cell Biology, 7(6), 456-462. https://doi.org/10.1038/nrm1925

Lichtenberg, D., Goñi, F. M., & Heerklotz, H. (2005). Detergent-resistant membranes should not be identified with membrane rafts. Trends in Biochemical Sciences, 30(8), 430-436. https://doi.org/https://doi.org/10.1016/j.tibs.2005.06.004

Shah, M. B., & Sehgal, P. B. (2007). Nondetergent isolation of rafts. Methods Mol Biol, 398, 21-28. https://doi.org/10.1007/978-1-59745-513-8_3